# Histone H1 facilitates restoration of H3K27me3 during DNA replication by chromatin compaction

Cuifang Liu[1,6], Juan Yu[1,6], Aoqun Song [1,2,6], Min Wang[1], Jiansen Hu[3], Ping Chen [1,4], Jicheng Zhao [1] ✉ & Guohong Li [1,2,5] ✉

During cell renewal, epigenetic information needs to be precisely restored to maintain cell identity and genome integrity following DNA replication. The histone mark H3K27me3 is essential for the formation of facultative heterochromatin and the repression of developmental genes in embryonic stem cells. However, how the restoration of H3K27me3 is precisely achieved following DNA replication is still poorly understood. Here we employ ChOR-seq (Chromatin Occupancy after Replication) to monitor the dynamic re-establishment of H3K27me3 on nascent DNA during DNA replication. We find that the restoration rate of H3K27me3 is highly correlated with dense chromatin states. In addition, we reveal that the linker histone H1 facilitates the rapid post-replication restoration of H3K27me3 on repressed genes and the restoration rate of H3K27me3 on nascent DNA is greatly compromised after partial depletion of H1. Finally, our in vitro biochemical experiments demonstrate that H1 facilitates the propagation of H3K27me3 by PRC2 through compacting chromatin. Collectively, our results indicate that H1-mediated chromatin compaction facilitates the propagation and restoration of H3K27me3 after DNA replication.

In eukaryotes, genomic DNA is hierarchically compacted into chromatin, with the nucleosome being its basic structural unit, which is formed by 146 base pairs of DNA wrapping 1.7 turns around an octamer of histones[1]. Chromatin states are dynamically modulated by numerous mechanisms to precisely regulate gene expression program. Histone post-translational modifications (hPTMs) are one of the most important epigenetic factors of transcription regulation. During DNA replication, parental histones, as well as newly synthesized unmodified histones, are deposited onto the daughter DNA molecule, which results in at least a twofold dilution of hPTMs[2]. In order to maintain cell identity, histone modifications should be precisely restored before the

next S phase[2]. Quantitative mass spectrometry studies confirmed the recycling of old histones with PTMs, which is the cornerstone of the restoration of hPTMs[3-5]. Furthermore, elegant studies using labeling of replicating DNA combined with next-generation sequencing (NGS) methods revealed that the histone modifications (H3K4me3, H3K36me3, H3K27me3, H3K79me3) are mostly preserved on daughter strands after DNA replication[6-9]. And the restoration of the histone PTM levels is not synchronous but exhibits mark- and locus-specific kinetics[7].

Using the CRISPR-biotinylation system to track parental histone deposition, a recent study revealed that only parental histones in

[1]National Laboratory of Biomacromolecules, CAS Center for Excellence in Biomacromolecules, Institute of Biophysics, Chinese Academy of Sciences, 100101 Beijing, China. [2]University of Chinese Academy of Sciences, 100049 Beijing, China. [3]Laboratory of RNA Biology, Institute of Biophysics, Chinese Academy of Science, 100101 Beijing, China. [4]Department of Immunology, School of Basic Medical Sciences, Beijing Key Laboratory for Tumor Invasion and Metastasis, Capital Medical University, 100069 Beijing, China. [5]Hubei Key Laboratory of Cell Homeostasis, College of Life Sciences, TaiKang Center for Life and Medical Sciences, Wuhan University, 430072 Wuhan, China. [6]These authors contributed equally: Cuifang Liu, Juan Yu, Aoqun Song. ✉e-mail: zjch@moon.ibp.ac.cn; liguohong@sun5.ibp.ac.cn

repressive chromatin domains could be largely inherited during DNA replication[10]. It is widely accepted that parental H3-H4 is recycled as an intact tetramer, which indicates that the associated modifications on H3/H4 are likely to be inherited[11]. Besides the deacetylation of H4K16, studies in yeast and higher eukaryotes have demonstrated that the repressive hPTMs, including H3K9me3 and H3K27me3, could be epigenetically inherited[12–14]. As a hallmark of facultative heterochromatin, H3K27me3 is catalyzed by polycomb repressive complex 2 (PRC2) to form broad repressive chromatin domains, which play an important role in silencing developmental and tissue-specific genes. Previous studies have revealed that the establishment/maintenance of H3K27me3 domains adopts a two-step mechanism[15]. First, JARID2 and MTF2 stably recruit PRC2 to "nucleation sites" and create H3K27me3-forming polycomb foci. Second, the preexisting H3K27me3 is recognized by the aromatic cage of EED, and thereby the enzymatic activity of PRC2 is allosterically activated to efficiently propagate H3K27me3 in cis or in far-cis via long-range contacts. In addition to the positive feedback model accounted for the inheritance of H3K27me3, the catalytic activity of PRC2 can also be regulated by other mechanisms, including the incorporation of EZH1/EZH2 and other accessory proteins, automethylation of its core subunits, different histone modifications, RNA and DNA sequences, and higher-order chromatin structures[15–19].

The linker histone H1 is an important chromatin regulator that compacts nucleosome arrays into 30 nm chromatin fibers[20]. Several studies have suggested that the linker histone H1 plays important roles in the establishment/maintenance of heterochromatin states[21–23]. Consistent with this hypothesis, knockout of H1c/d/e results in a significant reduction of H2AK119ub1 and H3K27me3 in mouse embryonic stem cells (mESCs) and reprograms epigenetic states in lymphocytes[23–25]. Using cryo-electron microscopy (Cryo-EM), the H1-compacted chromatin structure shows a left-handed double helix conformation with tetra-nucleosome as the structural unit that is stabilized by the pairwise interaction between Nth and (N+2)th nucleosomes[26]. Interestingly, the nucleosome-nucleosome pairing mechanism plays an important role in the RYBP/YAF2-PRC1-mediated propagation of H2AK119ub1 by bringing the neighboring nucleosome closer[24,27]. Therefore, we wonder whether a similar chromatin compaction/nucleosome pairing mechanism underlies the H1-dependent H3K27me3 establishment/restoration during cell renewal. In this study, we systematically investigate the restoration kinetic characterization of H3K27me3 after DNA replication using the quantitative ChOR-seq[7]. We also examine the effect of H1-mediated chromatin compaction on the propagation and restoration of H3K27me3 in vivo and in vitro. Our results strongly suggest that H1-mediated chromatin compaction facilitates the temporal-spatial restoration of H3K27me3 on newly synthesized chromatin, which might play an important role in cell fate determination and maintenance.

## Results

### H3K27me3 exhibits faster restoration at repressive regions post DNA replication

To investigate the restoration of H3K27me3 on nascent chromatin across the genome, we performed quantitative ChOR-seq[7,28] in embryonic stem cells (mESCs) and revealed the entire process of the re-establishment of H3K27me3 on nascent DNA. We prepared S-phase-synchronized mESCs, and short-pulse (15 min) labeled newly synthesized DNA with a thymidine analog (EdU) followed by sequential ChIP of H3K27me3[7]. EdU-labeled DNA can be biotinylated by click reaction and purified with biotin-streptavidin pull-down prior to next-generation sequencing (Fig. 1a). In this study, we performed cell cycle blocking with 2 mM thymidine 3 h after EdU labeling (T3) to prevent cells entering into the next S phase (Fig. 1a). To track the dynamic restoration of H3K27me3 on nascent DNA after DNA replication, we harvested the samples immediately after EdU labeling

(nascent chromatin, T0) and at a series of time points after labeling (mature chromatin, T2 to T6) (Fig. 1a). For the bioinformatic analysis, we first filtered EdU-labeled H3K27me3 regions from whole H3K27me3 peaks as the regions to calculate the restoration pattern/rate of H3K27me3 post-replication. We found that ~70% of H3K27me3 peaks were labeled by EdU in the synchronized mESCs, which is consistent with the previous findings that heterochromatins are mainly replicated during middle/late S phase[29,30] (Supplementary Fig. 1a, b). Previous studies have revealed that the restoration kinetics of histone PTMs post DNA replication greatly vary in a mark- and locus-specific manner[7]. In line with previous study[7], our results revealed that nascent H3K27me3 peaks were mainly restored at parental H3K27me3 peak regions (Supplementary Fig. 1d, e). To investigate the restoration kinetics of H3K27me3 in-depth post-replication, we categorized the restoration pattern of H3K27me3 into three clusters based on the timing when H3K27me3 reached the highest levels after replication using time series cluster analysis (Fig. 1b; Supplementary Fig. 1e). We observed that the H3K27me3 signals of cluster A which was defined as early restoration rapidly reached the maxima between T0 and T2. However, the H3K27me3 restoration of cluster B (intermediate restoration) and cluster C (late restoration) would take a much longer time (T2-T4; T4-T6, respectively) compared with cluster A (Fig. 1b, c). Cluster A to C accounted for 28.73%, 30.87% and 40.30% of H3K27me3-enriched regions, respectively (Supplementary Fig. 1f). To further characterize these clusters, we examined their genomic distribution on annotated genes defined by Refseq and found that the proportion of promoter regions in cluster C was greater than those in the other two clusters (Fig. 1d). Using epigenomic markers to annotate chromatin states, we found that multiple repressive histone modifications (including H3K9me2, H3K9me3 and H3K27me3), as well as histone H3, H4 and the linker histone H1e, were highly enriched in chromatin regions of cluster A. By contrast, active hPTMs (including H3K4me2/3 and H3K27ac) and H3.3 were enriched in cluster B and C. In line with this observation, clusters B and C also showed higher MNase-seq, DNase1-seq and ATAC-seq signals (Fig. 1e). To rule out bias caused by cell cycle blocking, we also performed ChOR-seq in asynchronous mESCs. Because a portion of late S phase cells rapidly pass through G2/M and the next G1 phase and enter into the next S phase after 3 h of EdU labeling, we collected samples at T0, T1, T2 and T3 time points after EdU labeling. In this condition, genome coverage analysis showed that EdU labeled ~80% of H3K27me3 peaks (Supplementary Fig. 1a, b). Venn diagram showed that the vast majority of EdU-labeled H3K27me3 peaks overlapped in synchronized and asynchronous mESCs (Supplementary Fig. 1c). Further analysis showed that the restoration pattern of H3K27me3 also can be categorized into three clusters that phenocopied the restoration of H3K27me3 in the synchronized mESCs (Supplementary Fig. 1g). These results suggest that H3K27me3 is more rapidly deposited at the compacted silent chromatin regions than the active chromatin regions post-replication.

### The restoration rate of H3K27me3 post-replication is highly correlated with repressive chromatin states genome-wide

To further explore the H3K27me3 genome-wide restoration kinetics in detail, we first analyzed the restoration patterns of H3K27me3 from cluster A to cluster C and found that they all followed the pattern of quadratic functions (Supplementary Fig. 1h, i). Hence, we defined the restoration rate (RR) as the average of the derivatives of the quadratic functions that we deduced (see "Methods" section for details) (Fig. 2a). Then we analyzed the restoration rate along H3K27me3 peak regions and found that the restoration rate was decreased gradually from the peak center to both sides, which indicates the heterochromatin domains may be formed by the propagation of H3K27me3 from its nucleation sites (Fig. 2b). Next, we calculated the H3K27me3 restoration rate per individual locus across the genome and grouped the H3K27me3-enriched regions into four clusters ("super-fast", "fast",

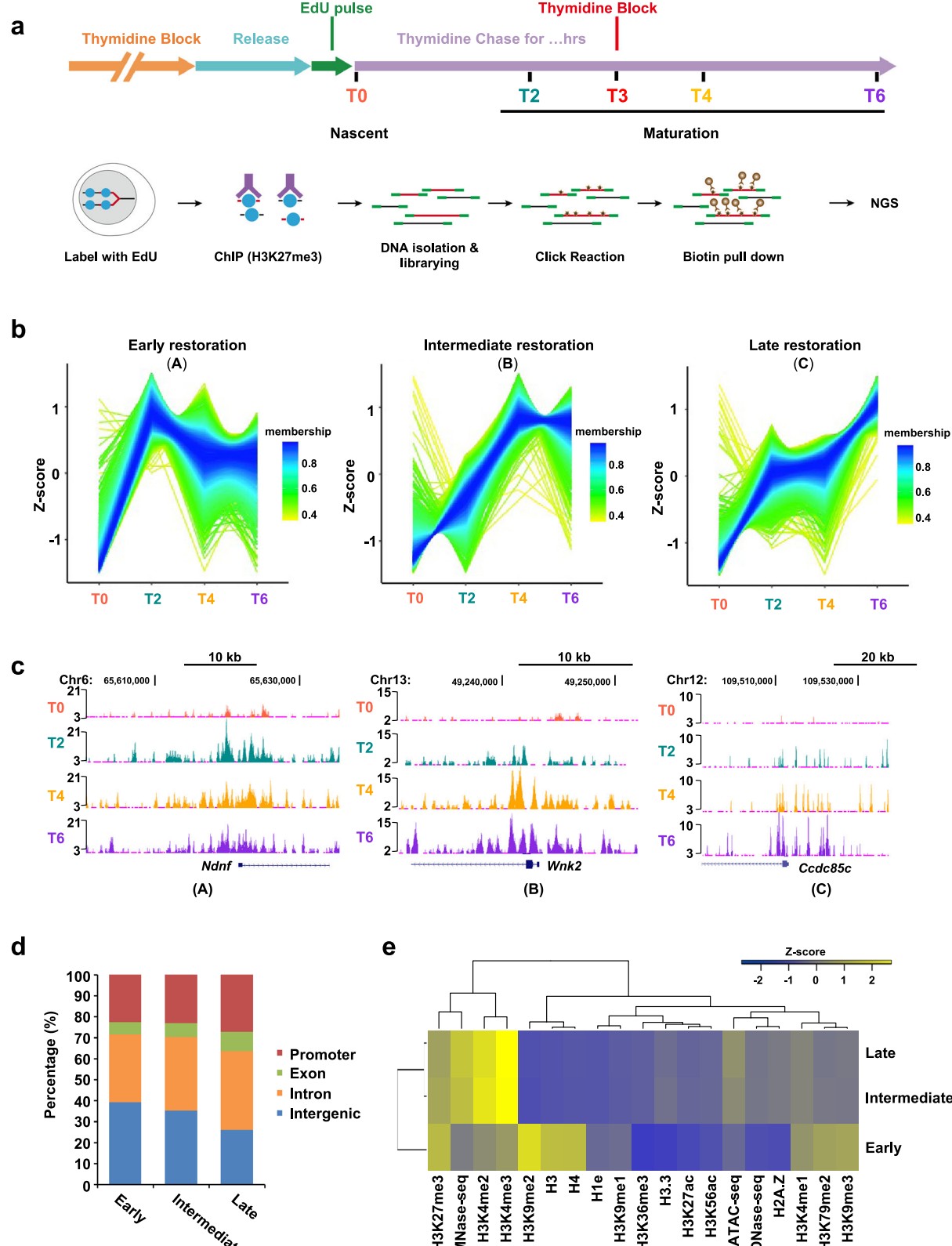

"slow" and "bottom-slow") (Fig. 2c). Analyzing the epigenetic features among these four clusters, we found that the restoration rate of H3K27me3 is highly positive-correlated with its own enrichments[7], the levels of linker histone H1e and H3K9me2/3 among all clusters, but anticorrelated with the levels of H3K27ac (Fig. 2d–g; Supplementary Fig. 2a–c). These results suggested that H3K27me3 restores faster at the condensed repressive chromatin regions. Further gene function

analysis revealed that these four clusters mainly enriched function terms involving cell morphogenesis, neural development and sensory organ development (Fig. 2h, i; Supplementary Fig. 2d, e), which is consistent with the functions of polycomb repressive complex in silencing development-related genes. To address the correlation between restoration rate and gene function in mESCs, we next collected the housekeeping, ES-specific, bivalent and tissue-specific gene

**Fig. 1 | H3K27me3 exhibits faster restoration at the dense repressive regions post DNA replication. a** Schematic of ChOR-seq workflow. Wild-type mESCs were released into the S phase from a 14 h thymidine block. Nascent chromatin was harvested immediately after EdU labeling (T0). mESCs were blocked with thymidine 3 h after EdU labeling (T3). Mature chromatin was collected at different time points (T2, T4, T6) following thymine chase for EdU labeling. The ChOR-seq assays in synchronized mESCs were performed twice. **b** Clusters of the restoration pattern of H3K27me3 at H3K27me3-enriched peak regions in wild-type mESCs using time series cluster analysis. A: Early restoration; B: Intermediate restoration; C: Late restoration. **c** Snapshot of tracks showed by UCSC Genome Browser from the three clusters of H3K27me3 restoration pattern in wild-type mESCs. **d** Genomic distribution analysis of early, intermediate and late restoration peaks of H3K27me3 according to Refseq annotations. **e** Hierarchical (Spearman rank) clustering of various histone modifications together with MNase-seq, DNase-seq and ATAC-seq signals at early, intermediate and late restoration peaks of H3K27me3. These data, except H1e, are from the public database; see Data Availability. Source data are provided as a Source Data file.

lists from publications[31–36] and divided H3K27me3 target genes into four categories. Intriguingly, we found that the clusters of "super-fast", "fast" enriched in tissue-specific and bivalent genes, but the "slow" and "bottom-slow" clusters enriched in mostly ES and housekeeping genes (Fig. 2j; Supplementary Fig. 2f). Consistent with these observations, the restoration rate of H3K27me3 is the lowest for housekeeping genes which have the lowest H3K27me3 level and the highest mRNA levels in mESCs (Fig. 2k–m). In addition, we got similar results in the asynchronous mESCs as that observed in synchronized mESCs (Supplementary Fig. 2g–n). Together, we hypothesized that the restoration rate of H3K27me3 after DNA replication is related to the functions of regulated genes with the fact that the rapid recovery of H3K27me3 in bivalent and tissue-specific genes and the super-slow restoration of this mark in housekeeping genes. Therefore, the restoration kinetics of H3K27me3 might play an important role in safeguarding cell identity by differentially regulating the expression of its target genes during cell division.

### H3K27me3 levels are regulated by the linker histone H1 in mouse embryonic stem cells

Previous studies have found that partial knockout of the linker histone H1 results in an obvious reduction of H3K27me3 and H2AK119ub1 in mESCs and/or lymphocytes[24,25]. In line with this, we here found that the H3K27me3 is rapidly restored at the dense chromatin regions compared with the open chromatin regions, suggesting that H1 may play an important role in regulating the post-replication restoration rate of H3K27me3 via modulating nascent chromatin structures. To address this hypothesis, we performed ChIP-seq analysis of H3K27me3 and H1e in mESCs, showing that H1e is colocalized with H3K27me3 across the genome (Fig. 3a; Supplementary Fig. 3a, b). Furthermore, we partitioned the H3K27me3-enriched regions into five clusters based on the enrichment of H1e and then analyzed the levels of H3K27me3 among these clusters (Fig. 3b). We observed a highly positive correlation between the levels of H3K27me3 and the enrichments of H1e, suggesting that the linker histone H1 may have a positive role in regulating H3K27me3 deposition. To this end, we generated the *H1c/d/e* triple knockout (H1-TKO) mESCs using CRISPR/Cas9, in which the total linker histone H1 proteins are reduced roughly ~50% of total linker histone H1 levels as reported previously (Fig. 3c; Supplementary Fig. 3c)[37]. Western blot analysis revealed that only H3K27me3 significantly decreased following H1-TKO among all the hPTMs we measured. However, the levels of SUZ12, a core component of PRC2, remained constant after H1 partial knockout (Fig. 3c). Consistently, ChIP-seq and ChIP-qPCR results showed that the enrichment of H3K27me3 level was significantly reduced in H1-TKO mESCs both at genome scale and at individual gene regions (Fig. 3d–f; Supplementary Fig. 3d, e). Moreover, the reduction of H3K27me3 levels was mainly observed at the 5' UTR and promoter regions of the genes (Supplementary Fig. 3f, g).

Furthermore, RNA-seq analysis revealed that H1-TKO resulted in 654 downregulated genes and 625 upregulated genes (Fig. 3g), which is consistent with previous findings that the reduction of H1 caused a specific set of genes misregulated, including both upregulated and downregulated[38]. Next, we found 43.81% (131) were upregulated among misregulated H3K27me3 targeting genes (299). Moreover, GO analysis showed that these upregulated genes highly related with developmental functions (Fig. 3h). Thus, we hypothesized that H1-dependent chromatin compaction would be essential for silencing these upregulated genes via promoting the restoration of H3K27me3 post-replication. Of note, 56.19% (168) of H3K27me3-trageted genes were downregulated in the H1-TKO mESCs, which may be caused by the indirect effects of H1-TKO. Alternatively, these results indicated that H1 might regulate gene transcription through other mechanisms, as previously reported, such as DNA methylation, H3K9me and HP1, and so forth[38]. Nevertheless, our results suggested that linker histone H1 plays a crucial role in regulating the establishment/maintenance of H3K27me3 at the dense heterochromatin that may be essential for silencing a subset of H3K27me3 target genes post-replication[23,37].

### H1-mediated chromatin compaction facilitates the restoration of H3K27me3 following DNA replication

EdU coverage analysis revealed that the usage of origins in wild type and H1-TKO mESCs are largely overlapped both in synchronized and asynchronous conditions (Supplementary Fig. 4a–d) that enable us to perform ChOR-seq in H1-TKO mESCs to study whether the post-replication restoration of H3K27me3 is tightly regulated by the H1-mediated chromatin compaction. To this end, we first carried out ChOR-seq in synchronized mESCs and found that the restoration pattern of H3K27me3 could also be categorized into three clusters based on the timing when H3K27me3 reached the highest levels after replication (Supplementary Fig. 4e, f). However, the proportion of early restoration cluster (A) significantly decreased in H1-TKO mESCs (Supplementary Fig. 4g). Hilbert curve showed that the H1-TKO resulted in a remarkable delay of the restoration of H3K27me3 at the level of whole chromosome (Fig. 4a). Furthermore, the restoration of H3K27me3 after DNA replication was significantly compromised in H1-TKO mESCs compared with WT mESCs both at transcription start sites (TSSs) and its peak regions (Fig. 4b–d; Supplementary Fig. 5a, b). Based on the definition of "restoration rate" in this study, we found that the restoration rate of H3K27me3 also decreased gradually from the peak center to both sides in H1-TKO mESCs, but the restoration rate at the peak center decreased to nearly 50% of that in WT mESCs (Figs. 2b, 4e). To identify genomic regions affected by H1-TKO, we compared the restoration rate of H3K27me3 in WT and H1-TKO mESCs among cluster A–D (clusters in Fig. 2c). Interestingly, we found that the restoration rate was significantly compromised in H1-TKO mESCs for cluster A–C, but not for cluster D, suggesting that H1-TKO greatly affected the restoration kinetics of H3K27me3 in the compacted chromatin regions (Fig. 4f). We also compared the restoration rate of H3K27me3 in housekeeping, ES, bivalent and tissue-specific genes in H1-TKO cells with those in WT cells. Consistently, the restoration rate of H3K27me3 was also significantly reduced in H1-TKO mESCs (Fig. 4g). Based on our ChIP-seq analysis, we next grouped H3K27me3 peak regions into four clusters (Q1 > Q2 > Q3 > Q4) according to the extent of H3K27me3 reduction in H1-TKO mESCs, revealing that histone H1e specifically enriched in Q1 group compared to other groups (Fig. 4h, i). We next examined the dynamic change of the restoration rate of H3K27me3 among these clusters in H1-TKO mESCs. As expected, the restoration rate of H3K27me3 in the Q1 cluster reduced more significantly than in other clusters in H1-TKO, which is in agreement with the dynamic change of H3K27me3 levels among clusters (Fig. 4j). Interestingly, we

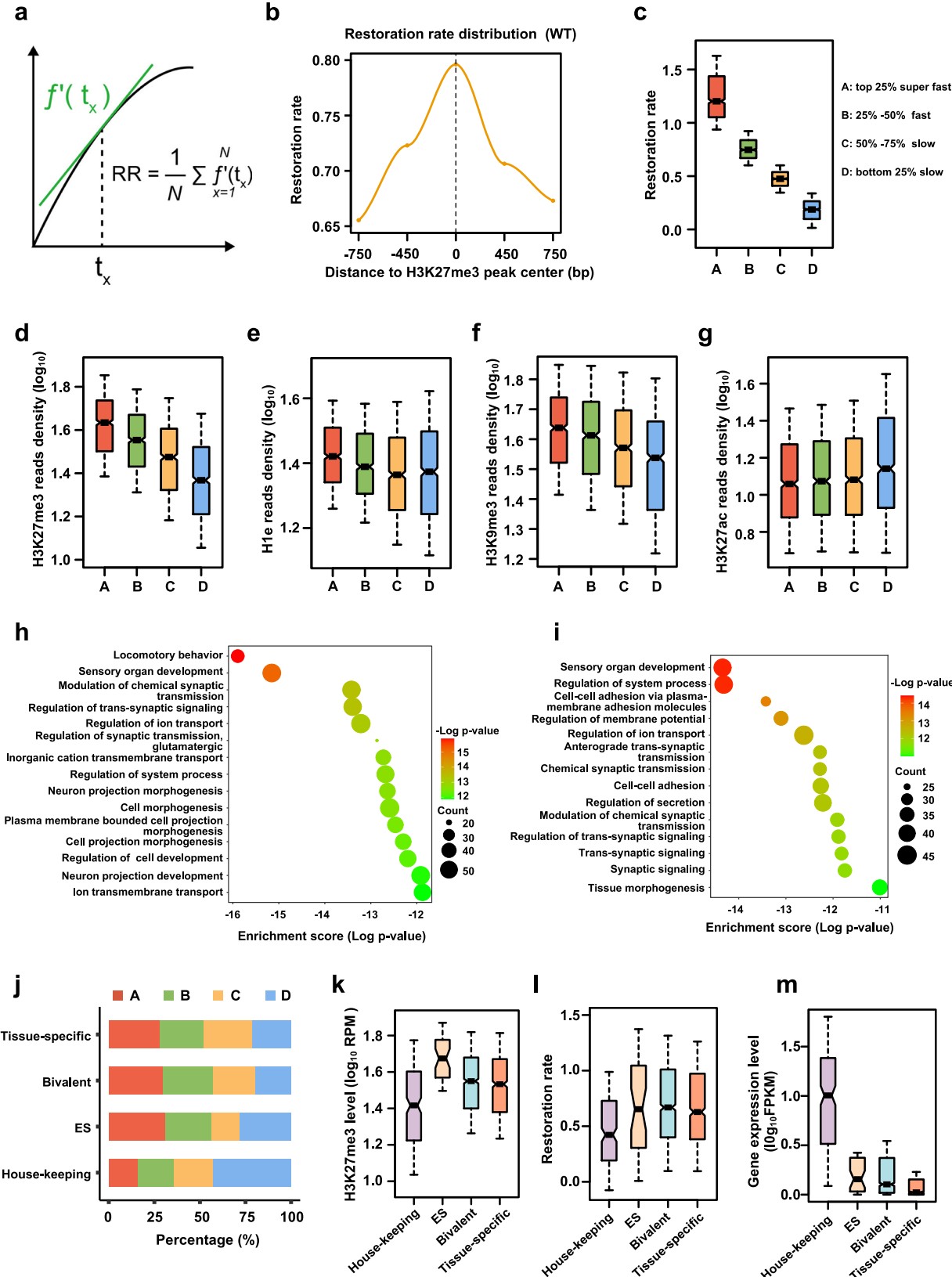

also observed the similar results under asynchronous condition (Supplementary Fig. 5c–h).

Accumulating studies suggested that H3K27me3 and PRC2 have a positive feedback loop in propagating H3K27me3[14], so we wondered whether the weakened restoration of H3K27me3 was caused by the reduced H3K27me3 levels in H1-TKO mESCs via alleviating the positive feedback loop. To solve this concern, we generated H1c/e inducible degradation mouse embryonic stem cell lines with stable-expressed 9×Myc-Tir1 (HA-AID-H1c/e cell lines). In this system, the degradation of HA-AID-H1c/e reached equivalent at ~12 h after IAA treatment; however, the H3K27me3 levels did not significantly change before ~24 h that endows us at least 12 h to investigate the

**Fig. 2 | The restoration rate of H3K27me3 is positively correlated with repressive chromatin states. a** Calculation formula for the restoration rate of H3K27me3, RR (Restoration rate). Please see details in the "Methods" section. **b** Average profiles of restoration rate of H3K27me3 in WT mESCs, plotted across ±750 bp centered at the peak summit of H3K27me3. Calculated using 1.5 kb windows with a 300 bp step; please see details in "Methods". **c** Boxplot showing the four clusters of H3K27me3 peaks labeled by EdU according to its restoration rate. **d**–**g** Box plots showing the average read densities of H3K27me3 (**d**), H1e (**e**), H3K9me3 (**f**) and H3K27ac (**g**) among cluster A–D. The ChIP-seq data are from the public database. Signals are quantitated using reads per kilobase per million (RPKM). **h**, **i** GO enrichment analysis of the genes in H3K27me3-enriched regions for clusters A (**h**) and clusters B (**i**). **j** Stacked bar chart showing the percentage distribution of clusters A–D covered genes among different gene functional groups. The genes were defined by EdU-labeled H3K27me3 at their promoter regions. **k**–**m** Boxplot showing the H3K27me3 level (**k**), restoration rate of H3K27me3 (**l**) and the gene expression levels (**m**) among different gene functional groups. RPM reads per million, FPKM fragments per kilobase of transcript per million fragments sequenced. **c**–**g** $n = 1046$ for A, $n = 1047$ for B, $n = 1046$ for C, $n = 1047$ for D. **k**–**m** $n = 202$ for housekeeping genes, $n = 39$ for ES genes, $n = 1378$ for bivalent genes, $n = 811$ for tissue-specific genes. The box plots (**c**–**g**; **k**–**m**) include the median line (median value indicated), the box denotes the interquartile range (IQR), whiskers denote the rest of the data distribution, and outliers are denoted by points greater than ±1.5 × IQR. Source data are provided as a Source Data file.

restoration kinetics with an equal H3K27me3 initial levels. To this end, we first cultured HA-AID-H1c/e cells in +IAA medium 6 h and then equally divided the cells into two sets: One set kept being treated with IAA (low HA-AID-H1c/e group), the other set was changed into fresh culture medium without IAA so that the H1c/e proteins are gradually restored to normal levels during 6 h (HA-AID-H1c/e restoration group) (Fig. 5a, b; Supplementary Fig. 5i). Subsequently, we performed ChOR-seq at T0, T1, T2 and T4 time points after IAA withdraw (Fig. 5c). Consistently, we found that the restoration of H3K27me3 is also significantly impaired following degradation of H1c/e (Fig. 5d–i). These results suggested that the H1-mediated chromatin compaction played a critical role in regulating the restoration kinetics of H3K27me3 following DNA replication.

### H1-mediated chromatin compaction facilitates the propagation of H3K27me3 by PRC2 via a nucleosome-nucleosome pairing mechanism

Previous studies have suggested that the enzymatic activity of PRC2 is allosterically activated when the aromatic cage of EED binds to H3K27me3, which allows H3K27me3 to efficiently propagate and form large heterochromatin domains via a positive feedback model[13]. Using in vitro histone methyltransferase assays (HMT), we monitored the deposition of H3K27me3 on various types of nucleosome/chromatin substrates. As expected, the activity of PRC2 was efficiently enhanced when using polynucleosome arrays as the substrate compared with mono-nucleosomes (Supplementary Fig. 6a). In addition, we found that the activity of PRC2 was significantly stimulated by the preexisting H3K27me3 marks (Fig. 6a). These results suggested that PRC2 adopted a positive feedback loop to propagate H3K27me3 marks along chromatin fibers. According to this model, we hypothesized that the propagation of H3K27me3 may be regulated by the distance between neighboring nucleosomes. Hence, we performed in vitro methyltransferase assays with H1-free and H1-compacted polynucleosome array substrates. Similar to the previous studies[16,23], the enzymatic activity of PRC2 was robustly stimulated when the chromatin was compacted by the linker histone H1 (Supplementary Fig. 6b, c). Moreover, time-course HMT assay results revealed that the H1-compacted chromatin greatly promoted the propagation of H3K27me3 along chromatin fibers both with and without preexisting H3K27me3 marks (Fig. 6b, c; Supplementary Fig. 6d). To rule out the stimulation was simply caused by the incorporation of linker histone H1, we further tested the activity of PRC2 on mono-nucleosomes with or without the linker histone H1. We found that PRC2 showed comparable enzymatic activity on mono-nucleosomes regardless of H1 addition (Supplementary Fig. 6e). These findings suggested that the propagation of H3K27me3 was promoted by the linker histone H1 mainly via chromatin compaction, which shortened the distance between neighboring nucleosomes. Subsequently, to get a direct observation of the propagation of H3K27me3 along chromatin fiber, we generated 12x-polynucleosome arrays via tetra-nucleosome ligation as previously described (Supplementary Fig. 6f–h). Using this system, we found that PRC2 was recruited and stimulated by

preexisting H3K27me3 and thereby led to the propagation of H3K27me3 from proximal to distal nucleosomes along the same chromatin fiber (Fig. 6d). Interestingly, the lateral propagation of H3K27me3 was robustly enhanced by H1-mediated chromatin compaction (Fig. 6d). To further confirm the propagation of H3K27me3 in vivo, we adopted a TetO/TetR targeting system in mESCs previously used in monitoring RYBP/YAF2-PRC1-mediated propagation of H2AK119ub1[24]. In this TetO/TetR targeting system, we observed a broad H3K27me3 domain was established around the 8×TetO sites when overexpressing HA-EED-TetR and FLAG-EZH2 or HA-EZH2-TetR and FLAG-EED in mESCs (Fig. 6e, f). However, when H1c/d/e was knocked out, the establishment of H3K27me3 domains was significantly compromised (Fig. 6e, f). Taken together, our results suggested that H1-compacted chromatin facilitates the lateral propagation of H3K27me3 along chromatin fibers via a positive feedback model and nucleosome-nucleosome pairing mechanism, which plays an important role in regulating the restoration kinetics of H3K27me3 and cell fate maintenance during the DNA replication.

## Discussion

Epigenetics is defined as inheritable information that is stored beyond DNA sequences. Histone modifications are important epigenetic mechanisms that are involved in almost all chromatin-related cellular events, including transcription, DNA replication, repair, etc. During DNA replication, parental and newly synthesized histones are deposited to leading and lagging DNA strands that result in at least a twofold dilution of histone modifications following the passage of replication forks. In order to maintain cell identity, the diluted histone modifications should be precisely restored before the next S phase. Previous studies have revealed that positional information of H3K4me3, H3K36me3, H3K79me3, and H3K27me3 is accurately restored on newly synthesized DNA via histone recycling, and the restoration of H3K4me3 and H3K27me3 levels also occurs via de novo methylation across the cell cycle with mark- and locus-specific kinetics[6–9,30]. In addition, the inheritance of H3K27me3 and H3K9me3 requires DNA-dependent recruitment and positive feedback mechanisms[14]. However, how the restoration of histone modifications on newly synthesized chromatin is temporally and spatially regulated is still poorly understood so far. Here we revealed that H1-mediated chromatin compaction plays an important role in promoting the restoration of H3K27me3 on nascent chromatin that may play important roles in maintaining transcriptional silencing of developmental and tissue-specific genes during the cell division in mESCs (Fig. 7).

### The restoration of H3K27me3 post replication and its potential roles in silencing developmental and tissue-specific genes in mESCs

The cell cycle is divided into a synthesis phase (S), a mitotic segregation phase (M) and two intervenient gap phases (G1 and G2) that precede S and M phases, respectively[39]. Previous studies have found that the initiation of differentiation mainly occurs in the G1 phase while the loss

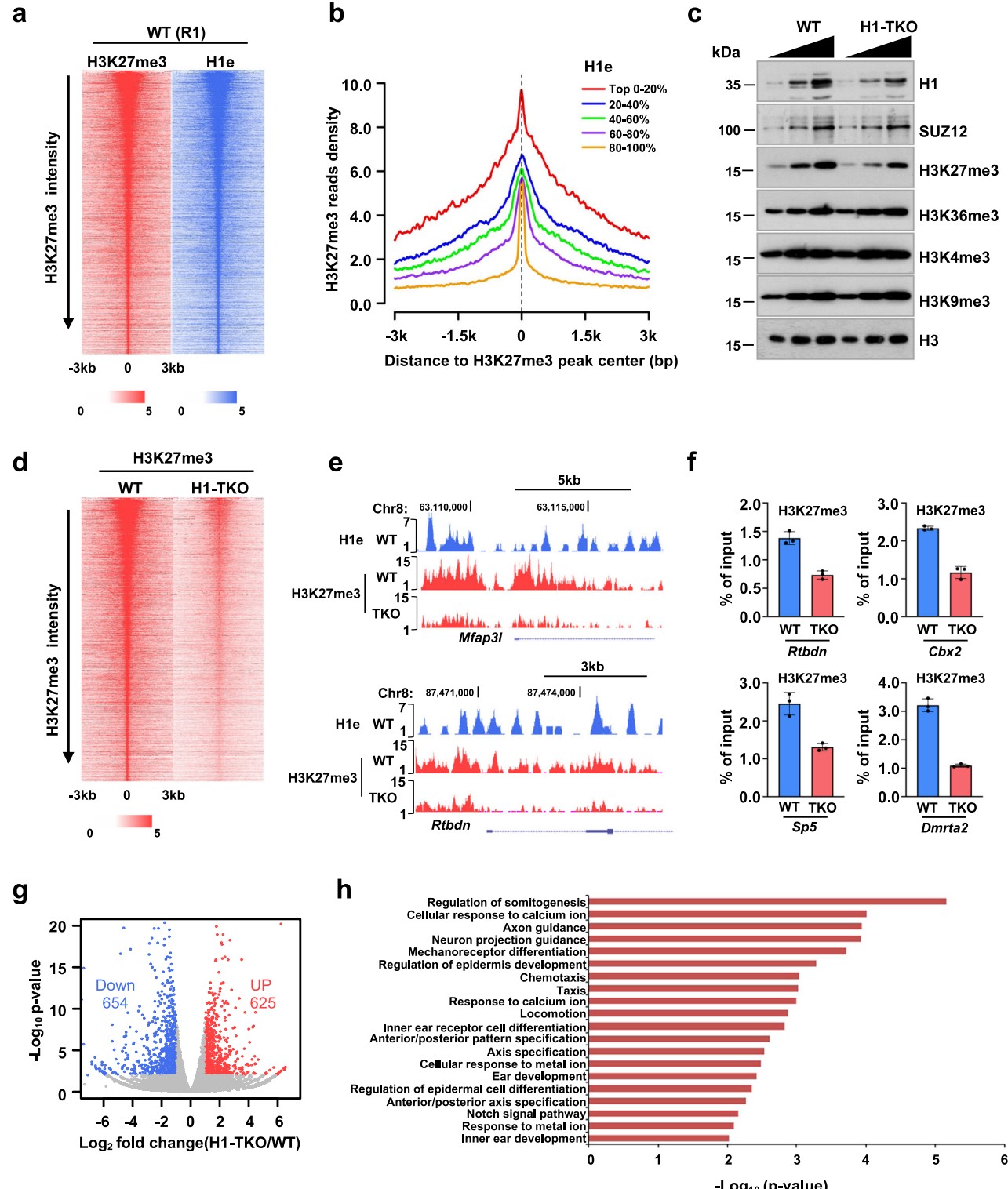

of pluripotency is subsequently achieved during S and G2 phases[40], suggesting that the epigenetic perturbation during the cell cycle is essential for cell fate determination. On the other hand, in order to achieve self-renewal of stem cells, daughter cells should faithfully inherit epigenetic information and chromatin states from mother cells during cell division. Inheritance of epigenetic information (e.g., DNA methylation and histone post-translational modifications) in daughter cells mainly occurs in the S and G2 phases[40] because of the chromatin disassembly ahead of replication forks during S phase[41–43] and the competition of transcriptional regulators with nucleosomes at newly

synthesized DNA post-replication[44]. To elucidate the principle of epigenetic inheritance in depth, in this study, we adopted ChOR-seq (Chromatin Occupancy after Replication) to monitor the dynamic process of the re-establishment of H3K27me3 on nascent DNA during DNA replication. We found that H3K27me3 is re-deposited onto its original position immediately after DNA replication, which is consistent with previous studies[7]. Due to the DNA replication-dependent dilution of H3K27me3, the T0 time point seems to have quite low enrichment. But this might not affect the relative restoration kinetics. Further analysis of H3K27me3 restoration patterns revealed that H3K27me3

**Fig. 3 | The linker histone H1 plays an important role in maintaining H3K27me3 levels in mESCs. a** Heat map analysis of H3K27me3 and H1e signals across H3K27me3-enriched regions (±3 kb), ranked from highest to lowest ChIP-seq signals of H3K27me3 in WT mESCs. Color intensity represents normalized and globally scaled tag counts. The H3K27me3 ChIP-seq in WT mESCs was performed twice. **b** ChIP-seq cumulative enrichment of H3K27me3 deposition centered at the peak summit of the H3K27me3, the H3K27me3 peak regions were clustered into five groups based on the levels of H1e. Signals are quantitated using reference-adjusted reads per kilobase per million (RPKM). **c** Representative immunoblots of H1, SUZ12, H3 and various histone modifications in WT and H1-TKO mESCs. H3 is shown as an internal reference. The results are representative of $n = 3$ biologically independent experiments. Triangles represent concentration of proteins from low to high. **d** Heat map analysis of H3K27me3 signals across its enriched peaks (±3 kb), ranked from highest to lowest ChIP-seq signals of H3K27me3 in WT and H1-TKO mESCs. Color intensity represents normalized and globally scaled tag counts. The H3K27me3 ChIP-seq in H1-TKO mESCs was performed twice. **e** Snapshot of tracks for ChIP-seq of H1e, H3K27me3 signals in WT and H1-TKO mESCs at *Mfap3l* and *Rtbdn* gene locus. **f** ChIP-qPCR of H3K27me3 levels in WT and H1-TKO mESCs ($n = 3$ biologically independent samples). The data represent means ± S.D. ChIP enrichments are normalized to the input of chromatin. **g** Volcano plot depicting mRNA changes for H1-TKO versus WT mESCs ($n = 2$ biologically independent samples). Statistical significance was determined by a two-sided Student's $t$-test. Adjustments were not made for multiple comparisons. **h** GO enrichment analysis of the upregulated genes in H3K27me3-enriched regions in H1-TKO mESCs. Source data are provided as a Source Data file.

restores faster at the dense chromatin regions that enrich with H1, H3K27me3, H3K9me2/3. In order to quantify the restoration rate of H3K27me3 at a genome-wide level, we defined a formula for calculating the restoration rate (RR) of this mark. Based on this model, we found that the "super-fast" and "fast" clusters of H3K27me3 restoration enrich with tissue-specific and bivalent genes, but the "slow" and "bottom-slow" clusters mostly enriched with ES and housekeeping genes in mESCs. Importantly, we also found that the restoration rate of H3K27me3 is negatively correlated with the mRNA levels of the genes among tissue-specific, bivalent, ES and housekeeping gene clusters, which might have a potential role in the post-replication gene silencing. It also supports the hypothesis that DNA replication provides a narrow time for cell fate determination changes by coordinating with intrinsic and extrinsic signals during the cell cycle.

### H1-mediated chromatin compaction plays an important role in regulating the restoration of H3K27me3 following DNA replication

The inheritance of H3K27me3 requires the recycling of parental histones that carry old hPTMs and the restoration of corresponding marks to pre-replication levels. Previous studies have shown that parental histones are retained and randomly positioned to newly synthesized DNA and then moved to their original positions during replication[44–50]. Similarly, a recent study has shown that repressed chromatin domains are preserved through cell division via the local re-deposition of parental nucleosomes[10,51]. Hence, these results suggest that the recycled hPTMs would serve as seeds to be propagated by the "read-write" mode at heterochromatin regions[14], which is consistent with previous findings that the transmission of H3K27me3-mediated repression during cell division needs DNA-dependent and -independent mechanisms[18,52]. Recently, the transmission of parental histones onto newly synthesized leading and lagging strands has been extensively explored[8,9,45,53]. However, the restoration kinetics of hPTMs on newly synthesized chromatin is still unclear.

H3K27me3 are deposited by PRC2. Biochemical and proteomic analyses revealed that mutually exclusive incorporation of accessory subunits divides PRC2 into two holocomplexes: PRC2.1 and PRC2.2[54–56]. PRC2.1 comprises a *polycomb*-like protein (PCL1/PHF1, PCL2/MTF2, or PCL3/PHF19) along with EPOP or PALI1/2. However, PRC2.2 comprises two zinc-finger-containing proteins (AEBP2 and JARID2)[56,57]. Genomic analyses have found that H3K27me3 distributes broadly and marks large facultative heterochromatin domains. Using an inducible H3K27me3-domain collapse and reconstitution system, a recent study reveals that PRC2 is stably recruited to a limited number of nucleation sites by JARID2 and MTF2 to form H3K27me3 seeds[15]. Subsequently, PRC2 is allosterically activated by the binding of EED with the H3K27me3 seeds and rapidly propagates H3K27me2/3 in cis and in far-cis via long-range contacts[15]. In our study, we found that the activity of PRC2 is greatly enhanced by preexisting H3K27me3 using in vitro HMT assays, supporting the idea that PRC2 is allosterically stimulated by its

products. In addition, cumulating studies have suggested that H2AK119ub1, noncoding RNA, H3K36me3 and CpG islands also play important roles in recruiting PRC2 to specific genomic regions[57]. In the nucleus, chromatin is hierarchically organized into higher-order chromatin structures that, in turn, serve as templates for histone-modifying enzymes. Several studies have found that the activity of PRC2 is activated by linker histone H1 on di- and polynucleosomes[16,58]; H1c/d/e-TKO and H1c/e-DKO lead to decreased H3K27me3 and increased H3K36me2 in lymphocytes[23,25], suggesting that higher-order chromatin structures play important roles in regulating PRC2 activity. Consistent with these observations, we found that H3K27me3 highly correlates with histone H1e across the genome, and H1-TKO results in a significant reduction of H3K27me3 in mESCs. In addition, the activity of PRC2 was further enhanced by H1 on polynucleosomes. These results suggested that the nucleosome-nucleosome pairing mechanism in H1-mediated chromatin compaction plays an important role in facilitating the propagation of H3K27me3 along chromatin fibers. Using ChOR-seq, we interestingly found that genomic regions that rapidly restore to their maxima of H3K27me3 after DNA replication enrich repressive hPTMs and H1e, suggesting the restoration of H3K27me3 may be regulated by the higher-order chromatin structure. Consistent with this, we found that the restoration of H3K27me3 after DNA replication is significantly compromised in H1-TKO mESCs compared with wild-type mESCs. Moreover, the restoration rate of H3K27me3 at the dense chromatin regions is greatly compromised by H1-TKO compared with that in the open chromatin regions. These results all support the conclusion that H1 plays a critical role in promoting the restoration rate of H3K27me3 during and after DNA replication via compacting chromatin structure, which obviously shortens the nucleosome distance via nucleosome-nucleosome pairing mechanism as described previously[24,27].

In sum, our study revealed the dynamic restoration of H3K27me3 in different functional regions of the genome and that H1-mediated chromatin compaction facilitates the restoration of H3K27me3 on newly synthesized chromatin which may contribute to remembering the developmental and tissue-specific genes in silencing states during the cell division in mESCs.

## Methods
### mESCs cell lines
All knockout and knock-in cell lines were generated by CRISPR/Cas9-mediated genome editing using pX260 vector. The guide sequence for *H1c/d/e* triple knockout mESCs is: 5′-CTTCTCCACATCGTACCCCG-3′ (conserved sequence for *H1c*, *H1d* and *H1e* genes in mouse). The guide sequence for HA-AID-*H1c* mESCs is: 5′-CCAGAGACTCATCATGTCTG-3′; The guide sequence for HA-AID-*H1e* mESCs is: 5′-AGCAGGCGCGG TCTCGGACA-3′. After transfection, we plated cells into three 10 cm culture plates with serially diluted density and screened with 1 μg/ml puromycin or 1 mg/ml neomycin. Ten days later, single-cell clones were picked out, and the genotypes were identified by PCR-coupled DNA sequencing. To evaluate the reduction of H1 in H1-TKO mESCs, we

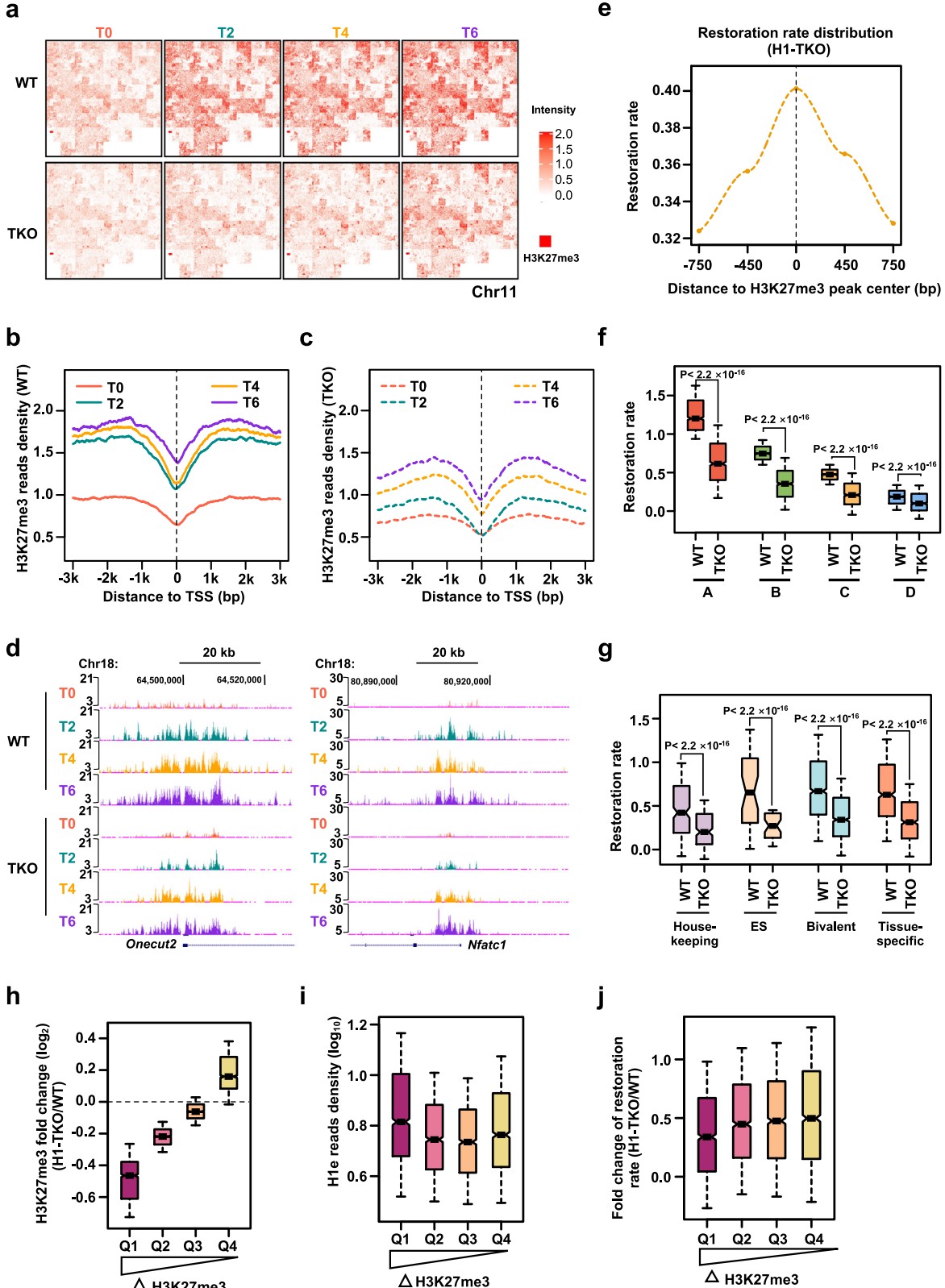

performed acid extraction for histones from wild-type and H1-TKO mESCs and evaluated the reduction of H1 in H1-TKO mESCs using Coomassie staining methods.

For generating 8×TetO knock-in mESCs, we directly fused the 8×*tetO* arrays with the human *CYP26A1* promoter that drives the expression of the *EGFP* gene. The guide sequence for CRISPR/Cas9 is: 5′-TCTGAACTGGCTGTTTAGTC-3′. To study the role of histone H1-

mediated chromatin compaction in promoting H3K27me3 propagation, HA-EZH2-TetR and FLAG-EED or HA-EED-TetR and FLAG-EZH2 were overexpressed in wild-type or H1c/d/e-TKO targeting mESCs, respectively. All mESCs were grown on gelatin-coated tissue culture plates in the presence of 1000 U/ml of mLIF (Millipore, USA) with Dulbecco's minimal essential medium (Millipore, USA) supplemented with 15% FBS (Ausbian, AUS), 100 mM nonessential amino acids,

**Fig. 4 | H1-mediated chromatin compaction facilitates the restoration of H3K27me3 following DNA replication. a** Hilbert curve of H3K27me3 ChOR-seq signals on chromosome 11 at different time points post DNA replication in WT and H1-TKO synchronized mESCs. Colored areas reflect the size and signal of H3K27me3-enriched domains. **b, c** Average profiles of H3K27me3 ChOR-seq signals at different time points post DNA replication centered at EdU-labeled gene TSSs in WT (**b**) and H1-TKO (**c**) synchronized mESCs. The ChOR-seq assays in synchronized mESCs were performed twice. Signals are quantitated using reference-adjusted reads per kilobase per million (RPKM). **d** Snapshot of tracks of H3K27me3 at different time points in WT and H1-TKO synchronized mESCs. **e** Average profiles of restoration rate of H3K27me3 in H1-TKO synchronized mESCs, plotted across ±750 bp centered at the EdU-labeled peak summit of H3K27me3. Calculated using 1.5 kb windows with a 300 bp step; please see details in "Methods". **f** Boxplot showing the comparison of restoration rate of H3K27me3 among clusters A–D in WT and H1-TKO synchronized mESCs ($n = 1046$ for A, $n = 1047$ for B, $n = 1046$ for C, $n = 1047$ for D). The *p*-values are calculated according to Wilcoxon signed-rank test (two-sided, $p < 2.2 \times 10^{-16}$). **g** Boxplot showing the restoration rate of H3K27me3 at housekeeping, ES, bivalent and tissue-specific gene cluster regions that enrich H3K27me3 marks (WT versus H1-TKO synchronized mESCs). $n = 202$ for housekeeping genes, $n = 39$ for ES genes, $n = 1378$ for bivalent genes, $n = 811$ for tissue-specific genes. The *p*-values are calculated according to Wilcoxon signed-rank test (two-sided, $p < 2.2 \times 10^{-16}$). **h** Four groups of H3K27me3-enriched regions classified according to dynamic changes of H3K27me3 after H1 TKO ($n = 1739$ for Q1, $n = 1739$ for Q2, $n = 1739$ for Q3, $n = 1740$ for Q4). **i** H1e reads density among Q1 to Q4 groups of H3K27me3 ($n = 1739$ for Q1, $n = 1739$ for Q2, $n = 1739$ for Q3, $n = 1740$ for Q4). Signals are quantitated using reads per kilobase per million (RPKM). **j** Boxplot showing the dynamic change of restoration rate of H3K27me3 in H1-TKO mESCs compared with that in WT mESCs among cluster Q1–Q4 ($n = 1739$ for Q1, $n = 1739$ for Q2, $n = 1739$ for Q3, $n = 1740$ for Q4). The box plots (**f**–**j**) include the median line (median value indicated), the box denotes the interquartile range (IQR), whiskers denote the rest of the data distribution, and outliers are denoted by points greater than ±1.5 × IQR. Source data are provided as a Source Data file.

0.1 mM β-mercaptoethanol, 2 mM L-glutamine, nucleosides and antibiotics (Millipore, USA).

## RNA preparation, RT-qPCR and poly(A) RNA-seq
The total RNA was purified with TRIzol reagent (Invitrogen) according to the manufacturer's instructions. Quantitative real-time PCR (qPCR) analyses were performed on an Applied Biosystems StepOne Plus Real-Time PCR system using the FastStart Universal SYBR Green Master (Roche). The primer pairs used for the qPCR experiments were the following:

Rtbdn: sense 5′-AGAGGACCATCTCAGCCTCA-3′, antisense 5′-GGATACTGAGGGCTGTTGGA-3′;

Cbx2: sense 5′-GCGAGCTCTTTCCAAAGCAG-3′, antisense 5′-AGGCTGCGAATCCAAAGTCA-3′;

Sp5: sense 5′-AGAGACTGTCTTGGCGGAAA-3′, antisense 5′-AACCGGAGTCAGAAAAGCAA-3′;

Dmrta2: sense 5′-CCTTGCAGGACCATGGTTCT-3′, antisense 5′-CATCTGCTAGGCACCCAGAG-3′. ChIP enrichment are normalized to the input of chromatin.

For poly(A)-RNA-Seq, DNA libraries were prepared according to the Illumina TruSeq protocol and sequenced using a HiSeq2000 system at Beijing Berry Genomics Co., Ltd.

## Western blotting and antibodies
The antibodies used in our study are: anti-HA (H3663, 1:2000 dilution) from Sigma; anti-H3 (ab1791, 1:5000 dilution), anti-H3K9me3 (ab8898, 1:2000 dilution), anti-H3K4me3 (ab8580, 1:2000 dilution), anti-H3K36me2 (ab9049, 1:2000 dilution), anti-H3K36me3 (ab9050, 1:2000 dilution), anti-H4K20me2 (ab9052, 1:2000 dilution) and anti-H4K20me3 (ab9053, 1:2000 dilution) from Abcam; anti-H1 (pAb) (61202, 1:2000 dilution) from Active Motif; anti-H3K27me3 (9733S, 1:2000 dilution) and anti-SUZ12 (3737S, 1:2000 dilution) from CST; anti-H3K9me2 (NB21-1072S, 1:2000 dilution) from NOVUS and anti-H3K4me2 (07-030, 1:2000 dilution) from Millipore; anti-V5 (P01L075, 1:2000 dilution) from Gene-Protein Link. The uncropped and unprocessed scans of all of the blots are shown in the Source Data file.

## Protein expression and purification
All histones were purified in BL21 (DE3) pLys *E. coli* cells that were transfected with pET3a-histones or pET28a-His-H1e plasmids, recombinant protein expression was induced by 0.5 mM IPTG for 4 h at 37 °C. For H2A, H2B, H3 and H4 histones, *E. coli* cell pellets were resuspended in 100 ml wash buffer (50 mM Tris-HCl, pH 7.5, 100 mM NaCl, 1 mM Na-EDTA, 1 mM benzamidine) and sonicated 200 cycles at 300 W. After centrifugation at 23,000×*g* for 20 min, inclusion bodies were washed with wash buffer + 1% Triton X-100 twice, and then washed with wash buffer twice. The inclusion bodies were finally resuspended in unfolding buffer (7 M guanidinium HCl, 20 mM Tris-HCl, pH 7.5, 10 mM

DTT) for resolving histone H2A, H2B, H3 and H4, respectively. For extracting His-tagged H1e, the inclusion bodies were resuspended in H1-unfolding buffer (1 M NaCl, 50 mM Tris-HCl, pH 7.5, 5 mM 2-Mercaptoethanol) and dialyzed into buffer A (50 mM Tris, pH 7.5, 500 mM NaCl), then sequentially purified with Heparin column and hydroxyapatite using a linear gradient (200–1000 mmol/L) $K_2HPO_4 + KH_2PO_4$ buffer (pH 6.8).

## Reconstitution of nucleosomes and chromatin fibers
For reconstitution of histone octamers, equimolar H2A, H2B, H3 and H4 histones were mixed and dialyzed against 2 L refolding buffer (2 M NaCl, 10 mM Tris-HCl, pH 7.5, 1 mM EDTA, 5 mM 2-Mercaptoethanol) overnight at 4 °C with at least two times changes of refolding buffer. The histone octamers were purified through Superdex 200 column (GE)[59].

Mono-nucleosomes and polynucleosomes were assembled using the salt-dialysis method[59]. Octamers and DNA were mixed in a buffer containing 10 mM Tris-HCl, pH 8.0, 1 mM EDTA and 2 M NaCl, and then the mixture was dialyzed against 450 ml of this buffer. Subsequently, the concentration of NaCl was gradually diluted to 0.6 M through pumping in 1×TE (10 mM Tris-HCl, pH 8.0, 1 mM EDTA). Finally, the nucleosomes were dialyzed with HE buffer (10 mM HEPES-KOH, pH 8.0, 0.1 mM EDTA). H1 was added before HE dialysis, and the mixture was dialyzed for an additional 3 h in 0.6 M NaCl buffer.

Heterotypic nucleosome arrays were generated as follows[24]. Briefly, 4×polynucleosome blocks were first constituted by the salt-dialysis method. Then the three chromatin blocks were joined using T4 DNA ligase (NEB, 5 U/μl) in 70 mM Tris-HCl (pH 7.5), 5 mM MgCl2, 1 mM ATP and 10 mM DTT. After 4 h incubation at 16 °C, the reactions were centrifuged at 17,000×*g* for 10 min at 4 °C. The pellets containing 12×polynucleosomes were re-dissolved in 1xTE buffer and dialyzed with 2 L 1×TE buffer overnight at 4 °C. For generating H1-compacted chromatin, ligated chromatin was dialyzed against 1 L 1×TE with 0.6 M NaCl and H1e was added for additional 3 h dialysis, then the buffer was changed into 1×TE via overnight dialysis. The uncropped and unprocessed scans of all of the blots are shown in the Source Data file.

## HMT assays
HMT assays were performed as reported previously with some modifications[59]. For details, the enzymatic reaction system includes 50 mM Tris-HCl (pH 8.5), 1 mM MgCl2, 4 mM DTT and 1 μg of nucleosome substrates. For quantifying H3K27me3 products, 0.5 μM S-adenosyl-L-[methyl-³H] methionine (³H-SAM) was added for each reaction, and enzymatic activity of PRC2 was titrated from 50 to 200 ng at 30 °C for 60 min. In the time-course experiments, 100 ng of PRC2 was used for each reaction, and they were incubated at 30 °C for the indicated times. The uncropped and unprocessed scans of all of the blots are shown in the Source Data file.

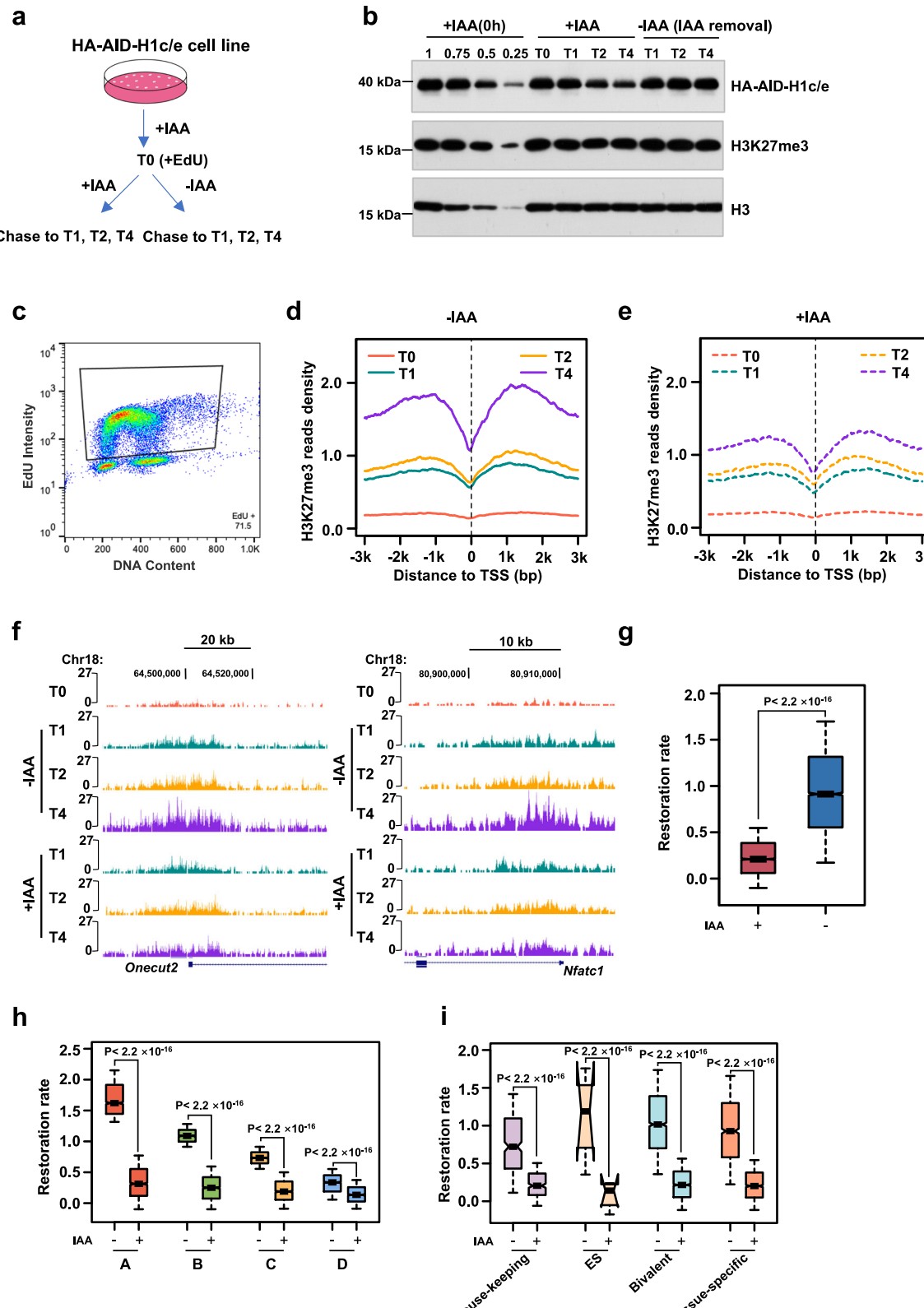

## ChIP-seq, ChOR-seq and replicated DNA isolation

For synchronized ChOR-seq set-up: R1 cells were synchronized at G1/S entrance with a single thymidine block (2 mM, 14 h) and then released the cells to a fresh medium for 2 h (middle S phase). The cells were pulse-labeled with 20 μM EdU (SANTA CRUZ, sc-284628) for 15 min. Immediately following the EdU pulsing, nascent chromatin samples (T0) were collected. The remaining cells were washed twice with cold

PBS and chased with fresh medium for different time lengths (T2-T6); we performed cell cycle blocking with 2 mM thymidine 3 h after EdU labeling (T3) to prevent cells entering into the next S phase. For asynchronized ChOR-seq set-up: R1 cells were pulse-labeled with 20 μM EdU for 20 min. Immediately following the EdU pulsing, nascent chromatin samples (T0) were collected. The remaining cells were washed twice with cold PBS and chased with fresh medium and

**Fig. 5 | H1-mediated chromatin compaction facilitates the restoration of H3K27me3 following DNA replication. a** Schematic showing the experimental flow of ChOR-seq in asynchronous HA-AID-H1c/e mESCs. The HA-AID-H1c/e mESCs were treated with IAA for 6 h and labeled with EdU 20 min immediately. Nascent chromatin was harvested immediately after EdU labeling (T0), mature chromatin was collected at different time points (T1, T2, T4) following chasing with IAA (+IAA) or without IAA (−IAA). The ChOR-seq assay was performed twice. **b** Representative quantitative-immunoblot showing HA-AID-H1c/e, H3K27me3 and H3 levels with or without IAA treatment at different time points. The results are representative of $n = 3$ biologically independent experiments. **c** Representative flow cytometry profiles of EdU-labeled cells in nascent samples. **d, e** Average profiles of H3K27me3 ChOR-seq signals at different time points post DNA replication, centered at EdU-labeled gene TSSs in −IAA (**d**) and +IAA (**e**) treated mESCs. (**d**) and (**e**) are quantitated using reference-adjusted reads per kilobase per million (RPKM). **f** Snapshot of tracks of H3K27me3 signal at different time points in −IAA and +IAA treated mESCs. **g** Boxplot showing the comparison of restoration rate of H3K27me3 labeled by EdU between −IAA and +IAA mESCs ($n = 6957$ for +IAA mESCs, $n = 6957$ for −IAA mESCs).

The $p$-values are calculated according to Wilcoxon signed-rank test (two-sided, $p < 2.2 \times 10^{-16}$). **h** Boxplot showing the comparison of the restoration rate of H3K27me3 among clusters A–D in −IAA and +IAA mESCs. In groups A, B and C, $n = 1739$ for +IAA mESCs, $n = 1739$ for −IAA mESCs; in group D, $n = 1740$ for +IAA mESCs, $n = 1740$ for −IAA mESCs. The $p$-values are calculated according to Wilcoxon signed-rank test (two-sided, $p < 2.2 \times 10^{-16}$). **i** Boxplot showing the restoration rate of H3K27me3 at housekeeping, ES, bivalent and tissue-specific gene cluster regions that enrich H3K27me3 marks (−IAA and +IAA mESCs). Housekeeping genes: $n = 338$ for +IAA mESCs, $n = 338$ for −IAA mESCs; ES genes: $n = 62$ for +IAA mESCs, $n = 62$ for −IAA mESCs; bivalent genes: $n = 2161$ for +IAA mESCs, $n = 2161$ for −IAA mESCs; tissue-specific genes: $n = 1338$ for +IAA mESCs, $n = 1338$ for −IAA mESCs. The $p$-values are calculated according to Wilcoxon signed-rank test (two-sided, $p < 2.2 \times 10^{-16}$). The box plots (**g**, **h**, **i**) include the median line (median value indicated), the box denotes the interquartile range (IQR), whiskers denote the rest of the data distribution, and outliers are denoted by points greater than ±1.5 × IQR. Source data are provided as a Source Data file.

---

incubated at 37 °C until collection at the indicated time point; we also performed cell cycle blocking with 2 mM thymidine 3 h after EdU labeling (T3) to prevent cells from entering into next S phase. *Drosophila* S2 cells were labeled with EdU (10 μM) for 39 h as spike-in chromatin samples.

The ChIP-seq and ChOR-seq protocol was adapted and modified as reported previously[7,8,60]. Briefly, R1 cells were immediately fixed in 1% formaldehyde for 15 min. Then, glycine was added to a final concentration of 0.125 M and the reaction was incubated for 5 min at room temperature. Fixed mESCs were resuspended in nuclei lysis buffer (50 mM Tris-HCl, pH 8.0, 10 mM EDTA, 1% SDS and protease inhibitors) and sheared to 300- to 500-bp-sized fragments. In parallel, *Drosophila* S2 cells were fixed, lysed, and sheared as described above. Then, 40 μg of sheared chromatin was mixed with 2 μg of *Drosophila* S2 chromatin and then diluted 10 times using ChIP dilution buffer (50 mM Tris-HCl pH 8.0, 0.167 M NaCl, 1.1% Triton X-100, 0.11% sodium deoxycholate and protease inhibitors). Diluted chromatin was pre-cleared with Dynabeads™ protein A/G 1 h before incubating with H3K27me3 (Cell Signaling Technology, 9733S) antibodies overnight at 4 °C. Then, antibody–chromatin complexes were captured with Dynabeads™ protein A/G and washed with RIPA150 (50 mM Tris-HCl pH 8.0, 0.15 M NaCl, 1 mM EDTA, 0.1% SDS, 1% Triton X-100, 0.1% sodium deoxycholate and protease inhibitors) once, RIPA500 (50 mM Tris-HCl pH 8.0, 0.5 M NaCl, 1 mM EDTA, 0.1% SDS, 1% Triton X-100, 0.1% sodium deoxycholate and protease inhibitors) twice, RIPA-LiCl (50 mM Tris-HCl pH 8.0, 1 mM EDTA, 1% NP-40, 0.7% sodium deoxycholate, 0.5 M LiCl and protease inhibitors) twice and 1×TE twice. Chromatin was eluted in 200 μl nuclei lysis buffer and de-crosslinked at 65 °C for 6 h, then digested with proteinase K (final concentration: 0.1 mg/ml) for 1 h digestion at 55 °C. DNA was extracted and then subjected to end repair, A-tailing, and adapter ligation using the NEB Ultra II DNA Library Prep kit (E7645) following the manufacturer's instructions for ChIP-seq assays. For ChOR-seq assays, EdU pulse labeled DNA was biotinylated under the following conditions: 15 mM Tris-HCl, 0.5 mM biotin-TEG-azide (Jena Bioscience), 0.1 mM CuSO₄ (Sigma-Aldrich), 0.5 mM THPTA (Sigma-Aldrich), and 10 mM sodium ascorbate (Sigma-Aldrich) for 45 min at room temperature. DNA fragments were purified using AMPure XP beads and resuspended in 1×TE buffer. Next, biotinylated DNA was purified using MyOne streptavidin C1 beads (Life Technologies) as follows: 6 μl beads (per sample) were balanced and washed with 1×BW&T buffer (5 mM Tris-HCl pH 7.5, 0.5 mM EDTA, 1 M NaCl, 0.05% (V/V) Tween-20) three times and then resuspended in 2×BW&T buffer. Streptavidin beads were then mixed 1:1 with biotinylated DNA and rotated for 45 min at room

temperature. Then, the beads were washed four times with 1×BW&T and twice with 1×TE with 0.05% Tween-20, then washed once with 10 mM Tris-HCl (pH 7.5) and resuspended in 10 μl ddH₂O finally. The beads were incubated at 98 °C for 10 min to perform 15 cycles of PCR amplification. The beads were separated with a magnetic rack and then PCR products in the supernatant were purified with AMPure XP beads (1×).

## Flow cytometer
For cell cycle analysis with propidium iodide (PI): cells were detached from the dish by trypLE™ express (Gibco) digestion and washed with 1×PBS. The cells were resuspended with 0.4 ml cold PBS, followed by adding 1 ml of cold 100% ethanol slowly on a vortexer. Then, the fixed cells were washed with 1 ml PBST and resuspended with 400 μl 10 μg/ml propidium iodide (Sigma) buffer supplemented with 20 μg/ml of RNase A (Thermo Fisher), and the reactions were incubated at room temperature for 30 min. Samples were stored on ice and protected from light until analysis by flow cytometry (20,000 cells per sample, BD FACS Calibur). Flow cytometry profiles were analyzed by FlowJo 10.

For EdU incorporation analysis: EdU pulsed cells were detached from the dish by trypLE™ express (Gibco) digestion and washed with PBS. The cells were resuspended with 0.4 ml cold PBS, followed by adding 1 ml cold 100% ethanol slowly on a vortexer. Then, the fixed cells were permeabilized in PBS with 0.25% Triton X-100 for 10 min at room temperature. Cells were spun down at 500×g for 5 min at room temperature and resuspended in 1×PBS for counting. Then, $5 \times 10^6$ cells were transferred to a new tube, spun down at 500×g for 5 min at room temperature, and resuspended in Click-IT reaction mix with the following conditions: 1×Click-IT buffer (Click-iT EdU Alexa Fluor 647 Imaging Kit, Thermo Fisher, C10340), and Alexa Fluor azide (1:1000), 2 mM CuSO₄, 1×Click-IT additive and 0.05% Triton X-100. Cells were incubated for 30 min at room temperature in the dark, then spun down and washed with 1 ml PBST four times. After that, cells were resuspended with 400 μl 10 μg/ml propidium iodide (Sigma) buffer supplemented with 20 μg/ml of RNase A (Thermo Fisher), and the reactions were incubated at room temperature for 30 min. Samples were stored on ice and protected from light until analysis by flow cytometry (20,000 cells per sample, BD FACS Calibur). Data were collected by CellQuest. Flow cytometry profiles were analyzed by FlowJo 10. The gating strategy used for cell cycle analysis is provided in Supplementary Fig. 7.

## Electron microscopy (EM)
Reconstituted chromatins were fixed with 0.4% glutaraldehyde (Fluka) in (10 mM HEPES, pH 8.0, 0.1 mM EDTA) on ice for 30 min and then mixed with a buffer containing 2 mM spermidine. The samples were loaded to the glow-discharged carbon-coated EM grids and incubated for 2 min. The grids were stepwise washed in 20 ml of 0%, 25%, 50%,

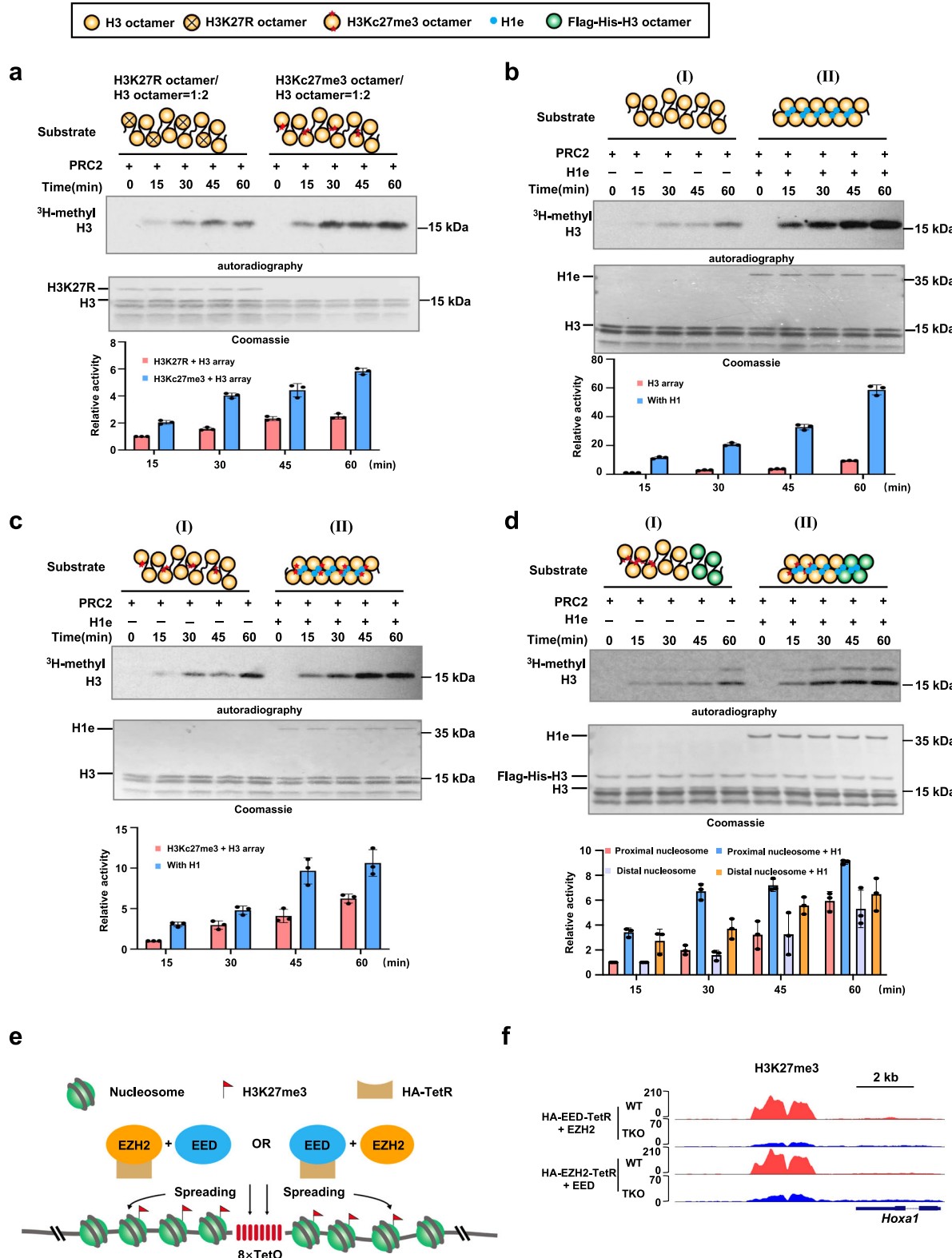

75%, and 100% ethanol solution for 4 min/each at room temperature, then air dried and shadowed with tungsten at an angle of 10° with rotation. The samples were evaluated under a FEI Tecnai G2 Spirit 120 kV transmission electron microscope.

**ChIP-seq analysis**

The previously published ChIP-seq data were downloaded from Sequence Read Archive BioProject under accession code: SRR925639,

SRR1044607, SRR925640, SRR1648485, SRR1130791, SRR925641, SRX837353, SRX8373532, SRX4386198, SRX111869, SRR2673297 SRR925647 SRX317656 SRR1799183 SRR1635424. The ChIP-seq generated in this study has been deposited in the GEO under accession code GSE192984.

All the ChIP-seq data were mapped to the *Mus musculus* genome (mm9) and *Drosophila melanogaster* genome assembly (dm6) using bowtie2[61] (version 2.2.5) with the default parameters. Low-quality

**Fig. 6 | H1-mediated chromatin compaction promotes the propagation of H3K27me3 via a nucleosome-nucleosome pairing mechanism. a–d** In vitro histone methyltransferase assays of PRC2 on different chromatin substrates, the signals of H3K27me3 products were detected by ³H-methyl autoradiography, quantitative analysis of ³H signals by liquid scintillation. The amount of substrate was detected by coomassie staining. The results are representative of $n = 3$ biologically independent experiments. The data represent means ± S.D. Schematics of the octamers used in chromatin assembly are shown above. In (**a**), the activity of PRC2 on polynucleosome substrates. Left, substrates contained a mixture of H3K27R and H3 octamers in a ratio of 1:2 (mixed before assembly). Right, H3K27R octamers were replaced by H3Kc27me3 octamers. In (**b**), the activity of PRC2 on polynucleosome and H1-compacted chromatin substrates. In (**c**), the activity of PRC2 on polynucleosome and H1-compacted chromatin substrates. Substrates contained a mixture of H3Kc27me3 and H3 octamers in a ratio of 1:2 (mixed before assembly). In (**d**), the activity of PRC2 on 12×polynucleosme substrates with different tetra-nucleosome blocks. H3Kc27me3, H3 and Flag-His-H3 tetra-nucleosomes were sequentially ligated. H1e was added to compact chromatin. **e** Schematic of TetO/TetR targeting system in mESCs. **f** Genome browser tracks of H3K27me3 peaks across the 8×TetO-integrated locus in WT and H1-TKO mESCs following the expression of HA-EED-TetR + FLAG-EZH2 or HA-EZH2-TetR + FLAG-EED. ChIP-seq assay was performed twice. Source data are provided as a Source Data file.

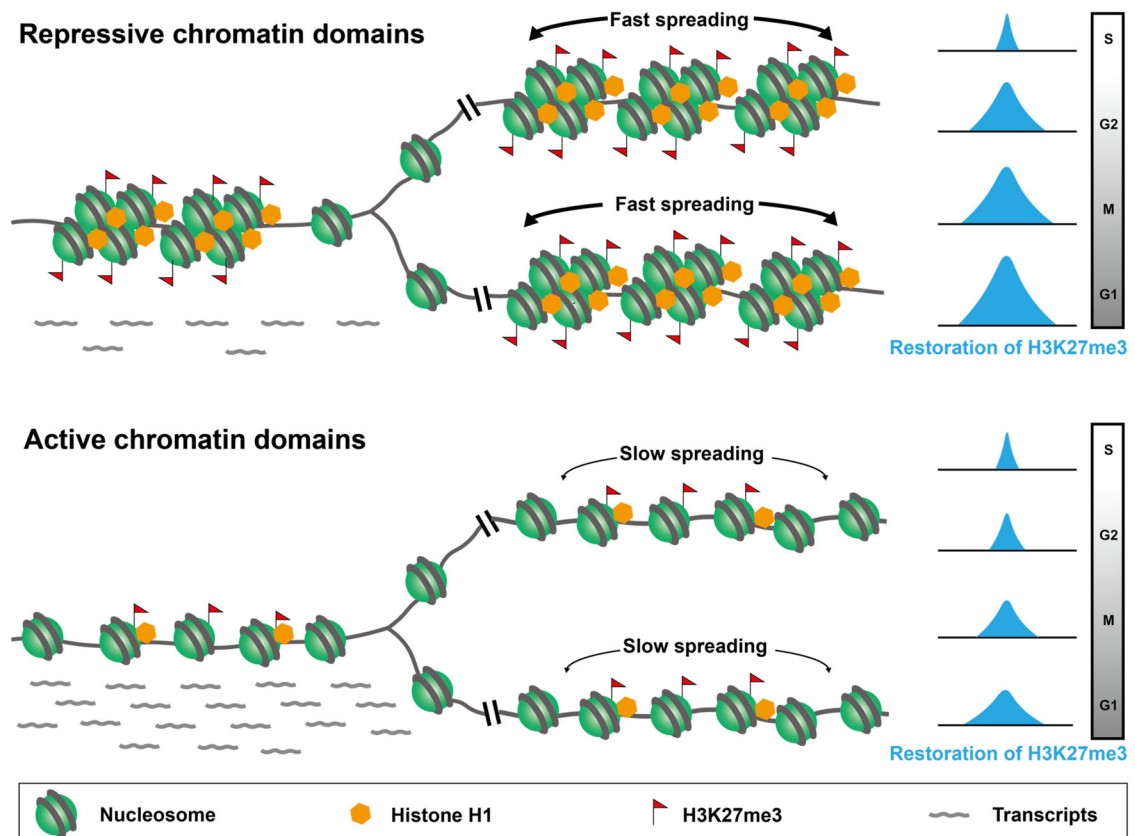

**Fig. 7 | Model of the post-replication restoration of H3K27me3 regulated by H1-mediated nucleosome-nucleosome pairing mechanism at dense heterochromatin regions.** During DNA replication, the levels of H3K27me3 on chromatin are diluted at least twofold by the incorporation of newly synthesized histones. In order to maintain the silencing of developmental and tissue-specific genes in mESCs, the linker histone H1 dominantly enriches at heterochromatin regions and compacts chromatin fibers via a nucleosome-nucleosome pairing mechanism that promotes the rapid restoration of H3K27me3 on the repressive genes immediately post-replication. On the contrary, the open chromatin structure with a low level of H1 retards the restoration of H3K27me3 at euchromatin regions that permit the expression of housekeeping genes throughout the cell cycle.

reads and PCR duplicates were removed by samtools (version 1.2.1) and retained the uniquely mapped tags. Enriched peaks were called using MACS[62] (version 1.4.2). The $p$-value cutoff for peak detection was $1e^{-5}$ with MACS14. Reads were normalized with fly reference as the previous study[63]. Peak overlapping was analyzed with Bed tools (version 2.17.0). HOMER[64] was used to annotate the peaks and count the read density at promoter and peak regions. Hierarchical clustering was performed using the unweighted pair group method, with the distance between profiles computed as Pearson's correlation coefficients by ggplot2. Heat maps were generated by Java Treeview[65]. Genome browser tracks were shown by UCSC Genome Browser[66]. Scatter plots, box plots and other plots were generated by R (http://www.r-project.org/) or Microsoft Excel.

### RNA-seq and gene expression analysis
The RNA-seq generated in this study has been deposited in the GEO under accession code GSE192984. The pair-end RNA-seq reads were mapped to the *Mus musculus* mm9 gene annotation model using Subread[67] with the default parameters, and only uniquely mapped reads were used for further analysis. The differential expression between samples was performed with featureCounts[68] and R package DESeq2[69]. Gene enrichment for one gene group was analyzed by Metascape[70].

### ChOR-seq bioinformatics analysis
The ChOR-seq data generated in this study have been deposited in the GEO under accession code GSE192984. All sequencing data

# Article

were mapped to the *Mus musculus* genome assembly (mm9) and *D. melanogaster* genome assembly (dm6) using bowtie2[61], with the default parameters. Low-quality reads and PCR duplicates were removed by samtools[71], and only uniquely mapped reads that map to a unique genomic location and strand were kept. Reads were normalized with fly reference[63]. The enriched regions (peaks) were detected using MACS[62]. The *p*-value cutoff for peak detection was $1e^{-5}$ with MACS14. Time series cluster analysis was performed by the TCseq package (https://bioconductor.riken.jp/packages/3.7/bioc/vignettes/TCseq/inst/doc/TCseq.pdf). Hilbert curves of H3K27me3 ChOR-seq data are visualized by the R package HilbertCurve[72]. Other analyses were performed as same as described for ChIP-seq data.

## Replicated regions definition and coverage calculation
The raw reads of EdU-labeled counts were generated over 1000 bp windows with a 100-bp step, and the probabilities were generated for each window by assuming a Poisson distribution. Windows with $p < 1 \times 10^{-4}$ after Benjamini–Hochberg correction were kept; overlapping windows were merged as replicated regions. H3K27me peaks, which overlapped with the replicated regions, were used for further analysis. The coverage is calculated by the proportion of H3K27me3 peaks labeled by EdU in all H3K27me3 peaks.

## Calculation of restoration rate
First, the normalized read frequencies for every enriched peak region were counted for each time-course point. Then, fitting quadratic function on the four values was performed for each peak region. Third, we selected n time points from T0 to Tx which correspond to the highest value of a quadratic function. Then, we counted the derivative for the n points, and the average of derivatives was used to represent the restoration speed. Here, we set n as 10000 and found that when n is greater than 1000, RT is relatively stable.

## Restoration rate (RR) distribution
Analysis regions were defined as ±750 bp centered at the summit of parental H3K27me3 peaks labeled by EdU. From the starting point, we slided a window every 300 bp and generated five windows. We calculated the RR for all five windows according to the calculation method of RR and then counted the average RR of all peaks in the same window.

## Reporting summary
Further information on research design is available in the Nature Portfolio Reporting Summary linked to this article.

## Data availability
ChIP-seq, ChOR-seq and RNA-seq data that generated in this study have been deposited in the GEO under accession code GSE192984. Previously published ChIP-seq data that were re-analyzed here are available in SRA under the accession codes: SRX7585130 (MNase-seq), SRR1703159 (ATAC-seq), SRR579162 (DNase-seq), SRR925639 (H3K4me1 ChIP-seq), SRR1044607 (H3K4me2 ChIP-seq), SRR925640 (H3K4me3 ChIP-seq), SRR1648485 (H3K9me1 ChIP-seq), SRR1130791 (H3K9me2 ChIP-seq), SRR925641 (H3K9me3 ChIP-seq), SRX8373531 (H3 ChIP-seq), SRX8373532 (H4 ChIP-seq), SRX4386198 (H3.3 ChIP-seq), SRX111869 (H2A.Z ChIP-seq), SRR2673297 (H3K27me3 ChIP-seq), SRR925647 (H3K36me3 ChIP-seq), SRX317656 (H3K27ac ChIP-seq), SRR1799183 (H3K56ac ChIP-seq), SRR1635424 (H3K79me2 ChIP-seq). Source data are provided with this paper.

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

## Acknowledgements

We thank Dr. Yanhui Xu for providing the PRC2 complex for the HMT assay. We are very grateful to Ting Yao for preparing the materials. This work was supported by the National Natural Science Foundation of China (31991161 and 32230020 to G.L., 32070604 and 32270614 to J.Z., 32100470 to C.L., 32022014 and 31871290 to P.C.), the Ministry of Science and Technology of China (2019YFA0508903 to J.Z., 2018YFE0203302 to P.C., 2017YFA0504202 to G.L.). This work was

also supported by the Beijing Municipal Science and Technology Commission (Z221100007022001) and a Howard Hughes Medical Institute (HHMI) international research scholar grant (55008737) to G.L., the Beijing Natural Science Foundation (7192027 to R.X.W. and J.Y.). All EM images were collected and processed at the Center for Bio-imaging, Core Facility for Protein Sciences, Institute of Bio-physics, and Chinese Academy of Sciences. All radioactivity experiments were performed at the radioactive isotope laboratory (Institute of Biophysics, CAS), with guidance from H.J. Zhang in handling radioactive materials.

## Author contributions

C.L. generated CRISPR-Cas9 edited cell lines, performed ChIP-seq, RNA-seq and biochemical experiments. A.S. performed ChOR-seq, Repli-seq and FACS experiments. J.Z. performed ChIP-seq for across the 8×TetO-integrated locus. J.Y. developed computational methods and performed computational analyses. M.W., J.H. and P.C. helped to discuss the project. G.L. conceived and supervised the project. J.Z. and G.L. analyzed the data and wrote the manuscript. All the authors read and commented on the manuscript.

## Competing interests

The authors declare no competing interests.
