## [Peer Review File · Nature Communications]

Histone H1 facilitates restoration of H3K27me3 during DNA replication by chromatin compactionEditorial Note: Parts of this Peer Review File have been redacted as indicated to remove third-party material where no permission to publish could be obtained.

REVIEWER COMMENTS

Reviewer #1 (Remarks to the Author):

In this manuscript, Liu and co-workers aim to address the role of histone H1 in restoration of K27me3 levels after replication. This is an important question and the ChORseq method is the appropriate tool to address this. The authors use ChORseq with spike-in correction to analyse restoration kinetics and conclude that repressed regions with high H3K27me3 levels restore most rapidly (consistent with findings in HeLa cells, Reveron-Gomez et al., 2018). They conclude that the restoration rate of H3K27me3 is pivotal for gene silencing, but no causative data are presented (repressed genes restore most rapidly, this does not mean that rapid restoration is required for silencing). The authors go on to show a positive correlation of the restoration kinetics to H1 occupancy, which reflects that there is more H1 at high occupancy K27me3 sites (known to restore most rapidly). The authors take advantage of H1-TKO to address the role of H1 in K27me3 post-replication restoration with the caveat that H3K27me3 levels are 2-fold reduced in this background (consistent with previous findings, Willcockson et al., 2021). The authors conclude that H1 governs H3K27me3 restoration kinetics. To support this model they show i) in vitro that H1 compaction of a chromatin array enhances PRC2 activity and K27me3 spreading (consistent with evidence on di-nucleosomes and cell-based analysis, Willcockson et al., 2021) and ii) that establishment of a K27me3 domain is less efficient in a H1 TKO background. Finally, they present a compelling model.

I overall find this story compelling, mainly because the model is highly consistent with earlier studies and most probably correct. It is well documented by the Skoultchi and Zhu labs that PRC2 activity and H3K27me3 occupancy is dependent on compaction and histone H1 (conclusions of Figure 5 and Figure 3). The novelty of this manuscript would be to demonstrate the relevance of this in restoration after replication. However, the conclusions on H3K27me3 restoration are not supported by the data as there are serious issues with the ChORseq experimental setup and data analysis. Also, a number of conclusions on causality are based on correlative observations. These serious concerns must be addressed before publication of the manuscript can be recommended.

Major points

The authors perform ChORseq in a synchronization set-up where mESCs are labelled after release into S phase 2hrs. There is no analysis of EdU labelled DNA and the genomic coverage, thus it is unclear what part of the genome they are tracking. If the aim is to address H3K27me3 dynamics across the genome and identify key parameters for restoration kinetics, the full genome must be covered by EdU labelling. The authors therefore would need to repeat the ChOR-seq experiments by labelling asynchronous cells.

There is little information provided about the data analysis, but it appears that the ChORseq analysis is done without filtering for replicated regions, which is essential in a synchronization set up. By including regions that are not covered by the EdU labelling, conclusions will be based on changes in background signal (i.e, regions that have not been replicated in a high proportion of the cells or not replicated at all in the 15min EdU pulse during early/mid S).

The use of a synchronization setup is also problematic for the comparison between WT and H1 TKO cells, as it assumes that the same set of regions will be replicating during the EdU pulse in early/mid S in both cell lines. This problem would be circumvented by doing ChORseq in asynchronous cells.

The information provided on data analysis is very limited. The authors refer to the software but do not specify how they normalize the signal, how the different units are computed (for example, read density) or how peaks are defined (from ChIP, from ChORseq...). Without this information it is not possible to evaluate the conclusions based on quantitative differences, for example. They do also not

specify how the different gene categories are defined; i.e. H3K27me3 peaks are generally not found on housekeeping promoters so what are they looking at?

The major conclusions of Figure 3 and Figure 5 were published previously by Fan et al., 2005 and Willcockson et al., 2021, reducing overall novelty of the work. H3K27me3 restoration kinetics was also previously shown to be fastest at sites with high K27me3 and PRC2 binding (Reveron-Gomez et al., 2018). It would be advisable to cite these publications when concluding on experiments that validate or corroborate previously findings (even if the papers are cited elsewhere in the manuscript).

Given that the K27me3 level is 2-fold reduced in the H1 TKO cells, it not clear whether the lower K27me3 or lack of H1 is responsible for the reduced restoration kinetics. An inducible system with similar baseline prior to replication would be preferable. The authors could also compare in WT cells restoration kinetics of regions with similar K27me3 levels and different H1 density.

There are conclusions on causality based on correlations mainly concerning the importance of restoration rates for gene regulation.

Minor points

It is unclear what data was produced in this manuscript and what data comes from published datasets. There are no accession numbers or references to any published data.

The authors refer to different genomic regions without defining them or providing any information on their overlap, e.g. they classify first restoration kinetics of peaks and then calculate the restoration kinetics to pre-defined promoters (ES, housekeeping, bivalent etc.). It is also not clear if these promoters have H3K27me3.

Details missing in the figures and figure legends, mostly units and data source.

Details missing in the methods, e.g there is a second thymidine addition in the schematic in Fig.1 but this is not clear from the methods.

Reviewer #2 (Remarks to the Author):

Mechanisms of propagation and maintenance of epigenetic regulation across cell cycle are of significant interest, both because of fundamental importance to developmental biology, and due to emerging understanding that many human disorders are driven by misregulation of gene expression. The idea that repressive chromatin modifications are particularly dependent on the "inheritance" mechanisms during S phase has been developed in significant detail by many laboratories, including Reinberg (e.g., Reinberg and Vales, Science 2018, and Escobar et al., Cell 2019), Grewal (Holla et al., Cell 2020), Crabtree (Chory et al., Mol Cell 2019) and others. Likewise, emerging interest in the function of linker histone H1 is highlighted by recent publications from Skoultchi and Melnick laboratories (Willcockson et al., Nature 2021 and Yusufova et al., Nature 2021) demonstrating, among others, the effect of H1 loss on core histone modification landscape. While previous results imply that PRC2 methyltransferase complex is stimulated by H1 incorporation in vitro (Martin et al., JBC 2006), mechanistic in vivo evidence is incomplete. Skoultchi and Fan groups have previously demonstrated that loss of H1 in mES cells impairs differentiation [e.g., Zhanmg et al., Plos Genet 2012] but given the complexity of the phenotypes, specific relationship between H1 incorporation and PRC2-driven

H3K27 methylation must be studied further.

The current study attempts to tackle this question using several experimental approaches. First, the authors study H3K27me dynamics in post-replication chromatin using ChOR-Seq. Second, they generate and characterize H1 triple-knockout cell line. Third, they perform several biochemical assays to assess the impact of H1 incorporation on PRC2 function in vitro.

While the paper is undoubtedly timely and the question under investigation is of broad interest and significance, I have several concerns about both the experimental setup and outcomes.

1. I have some concern about the ChOR-Seq setup. Replication is not uniform across the S-phase - with repressive chromatin generally contained in late-replicating domains. If the cells are indeed synchronized at the onset of S phase (as stated at line 187 of supplementary material - although Fig S4A argues that quite a few cells are in the middle of S?), and S phase takes several hours to complete, I would expect that early-replicating domains, predominantly found in active chromatin, would be the first to incorporate EdU - and more delayed-replicating domains, located in K27me-repressed (and ultimately K9me-repressed) chromatin would engage in replication much later. A recent paper by Glibert group (Zhao, Sasaki and Gilbert, *Genome Biol* 2020) has mapped several classes of initiation zones in mES cells, with roughly 1500 initiating early, and ~1000 initiating later in the S-phase. At the very least, I'd like to know how do these correspond to ChOR-Seq data in the current manuscript? Given that, to the naked eye, most robust changes in K27me3 happen between T0 and T2 (Figure S1B, and Figure 4A, B, D), I'd like to know more about these regions (it is somewhat counterintuitive that K27me3-rich regions are replicated within first two hours of release).

2. Perhaps even more puzzling, looking at the data in Fig S1B, I DO NOT see three distinct classes of K27me3 restoration. Even though the peaks are stated to be scaled by parental K27me3 density (or peak height?), they all look virtually identical both in their dynamics (change from T0 to T6) and original peak intensity (top to bottom in "Parental" heatmap. Given that much of the paper is based on interpretation of these data, it's critical to show that there are indeed distinct restoration clusters. Figure 1B looks like it, and perhaps some of the discrepancy is due to the scale differences (S1B is scaled -3 to 3, and 1B is scaled -1 to 1). Yet I do not see how clusters A, B and C are derived from raw data shown in S1B.

3. As overall steady-state levels of K27me3 are different between the two cell lines used in the study (see fig. 3C, D - and previously reported e.g. by Willcockson et al., *Nature* 2021), and pre-existing K27me3 levels determine the rate of K27me3 spreading, I am not convinced that this system allows the authors to unequivocally state that H1 loss is directly affecting the rate of spreading. We know that H1 appears to stimulate PRC2 even in di-nucleosomal substrates (first reported by Martin et al., *JBC* 2006) - I think the jump to compaction as mechanism is intriguing and very plausible, but is not fully supported by data. Further, comparing data in 2C-G, I am left with the impression that the greatest predictor of K27me3 restoration rate (Fig. 2C) is K27me3 read density (Fig 2D) - not the H1e read density (Fig 2E).

4. The RNA-Seq in H1 TKO cells is a bit puzzling given almost identical number of genes going up and down. I would have expected more genes being upregulated given how dramatically H3K27me3 appears to be affected (and given what we know about H1 as a repressor). The GSEA plots are not informative as shown. Do the upregulated genes encompass sets found within cluster A? What are the downregulated genes?

5. The previous concern could be partially alleviated by a more robust distinction between recruitment and spreading. If H1 is indeed affecting the spreading dynamics, then the K27me3 peak height should remain roughly similar - but the breadth is expected to be reduced dramatically. At the very least, I'd expect a faster dropoff (such that K27me3 peaks in WT would essentially represent "obtuse angles", and in TKO would be more akin to "acute angles") from the peak center to the edges. The authors do the right experiment in Fig 5E-F, but looking at the data, I don't see a difference in K27me3 peak

shape - only an overall decrease in H1 TKO background - therefore, I still can't parse out whether H1 loss causes defect in "recruitment" or "spreading" in vivo.

6. Several essential controls are missing:

a. While the original H1 TKO lines generated by Skoultchi lab have been extensively characterized and show about 2-fold reduction of total H1 levels, the newly generated lines reported in the paper require additional characterization. What is the genomic landscape of CRISPR-generated lines? Did Cas9 editing create large deletions or small substitutions? How were the clones isolated (or are these non-clonal lines?) What are the levels of H1 (can be estimated by HPLC, or even by Coomassie staining, since H1 isoforms have an obvious migration pattern - this is critical since H1 genes are dosage-compensated!)

b. are K27me3 levels uniform in these cells? (metastable mES cells grown in serum/LIF media have significant variability in H3 K27me3 levels - are the K27me3 "low" cells cycling faster or slower? (basically, which cell state contributes most signal to the ChOR?). I am not sure "original" TKO cells from Skoultchi lab were ever tested for K27me3 heterogeneity, and interestingly, the original report (Fan et al., Cell 2005) did not report significant changes in K27me - but it seems like an important question. Perhaps analyses similar to Fig S4A can be done to assess the K27me3 levels during the cell cycle?

c. To add to Fig S4A, I'd like to see more detailed quantification of cell cycle progression in these cells. E.g., it appears that T4 plots are a bit different - is it an outlier or part of the phenotype?

d. In the biochemical experiments reported, the authors need to show that H1e expressed and purified is full length. H1s are notoriously hard to purify without degradation, and simple His-tag system may be insufficient (several workarounds have been reported in the literature, including dual tags - Osunsade et al., Biochemistry 2019). Coomassie gels shown in Fig. 5 should be cropped above 30 kDa size to account for H1.

7. Throughout figures (e.g., fig 1B, E, fig 3 A, D - and several supplementals) labels in heat map color scales are missing. Whether these are z-scores, log2 fold change, log2 of read counts, or other value - should be labeled as such.

There is no question that the paper is timely and of broad interest, yet given that in vivo experiments shown have several significant caveats, and in vitro experiments largely replicate known results (Martin et al. 2006 and Willcockson et al., 2021), I am not fully convinced the paper is ready for publication until the authors are able to disentangle defect in K27me3 spreading from overall decrease in K27me3 observed in their system at steady state.

Reviewer #3 (Remarks to the Author):

In the manuscript "Histone H1 facilitates restoration of H3K27me3 during DNA replication by chromatin compaction", Liu et al. reported that H1 facilitates the rapid post-replication restoration of H3K27me3 on repressed genes using ChOR-seq. The authors also showed that H1 facilitates the propagation of H3K27me3 by PRC2 in vitro and that the H3K27me3 restoration on nascent DNA is compromised in H1-TKO mESCs. Overall, the experiments were well performed, and results were clearly presented. ChOR-seq provided specifics on the rapid restoration of H3K27me3 following DNA replication, although H1 facilitating H3K27me3 propagation is somewhat expected given the substrate preference of PRC2 for H1 containing nucleosomes (Martin C 2006). The drastic effects by H1-TKO in mESCs observed in this study (dramatic gene expression changes (>1200 genes) and H3K27me3 ChIPseq), however, are in stark contrast to findings from previous studies. It has been well documented that H1-TKO produces limited expression changes /negligible changes in H3K27me3 ChIPseq in undifferentiated mESCs but induces dramatic changes in more specialized/differentiated cells (Fan 2005, Zhang 2012, Geeven 2015, Encode, Willcockson 2021, Yusufova 2021, etc.). How many mESC lines were analyzed and how were these H1-TKO mESCs by CRISPR/Cas9 characterized? It's not clear why the authors used H1-TKO mESCs by CRISPR/Cas9 when only found H1e enrichment

with H3K27me3 and in cluster A in mESCs. The significance of H1e in rapid restoration of H3K27me3 in vivo remains to be addressed.

Reviewer #1 (response to the reviewer):

In this manuscript, Liu and co-workers aim to address the role of histone H1 in restoration of K27me3 levels after replication. This is an important question and the ChORseq method is the appropriate tool to address this. The authors use ChORseq with spike-in correction to analyse restoration kinetics and conclude that repressed regions with high H3K27me3 levels restore most rapidly (consistent with findings in HeLa cells, Reveron-Gomez et al., 2018). They conclude that the restoration rate of H3K27me3 is pivotal for gene silencing, but no causative data are presented (repressed genes restore most rapidly, this does not mean that rapid restoration is required for silencing). The authors go on to show a positive correlation of the restoration kinetics to H1 occupancy, which reflects that there is more H1 at high occupancy K27me3 sites (known to restore most rapidly). The authors take advantage of H1-TKO to address the role of H1 in K27me3 post-replication restoration with the caveat that H3K27me3 levels are 2-fold reduced in this background (consistent with previous findings, Willcockson et al., 2021). The authors conclude that H1 governs H3K27me3 restoration kinetics. To support this model they show i) in vitro that H1 compaction of a chromatin array enhances PRC2 activity and K27me3 spreading (consistent with evidence on di-nucleosomes and cellbased analysis, Willcockson et al., 2021) and ii) that establishment of a K27me3 domain is less efficient in a H1-TKO background. Finally, they present a compelling model.

I overall find this story compelling, mainly because the model is highly consistent with earlier studies and most probably correct. It is well documented by the Skoultchi and Zhu labs that PRC2 activity and H3K27me3 occupancy is dependent on compaction and histone H1 (conclusions of Figure 5 and Figure 3). The novelty of this manuscript would be to demonstrate the relevance of this in restoration after replication. However, the conclusions on H3K27me3 restoration are not supported by the data as there are serious issues with the ChORseq experimental setup and data analysis. Also, a number of conclusions on causality are based on correlative observations. These serious concerns must be addressed before publication of the manuscript can be recommended.

Major points

1. The authors perform ChORseq in a synchronization set-up where mESCs are labeled after release into S phase 2hrs. There is no analysis of EdU labelled DNA and the genomic coverage, thus it is unclear what part of the genome they are tracking. If the

aim is to address H3K27me3 dynamics across the genome and identify key parameters for restoration kinetics, the full genome must be covered by EdU labelling. The authors therefore would need to repeat the ChOR-seq experiments by labelling asynchronous cells.

Response: We thank reviewer for this suggestion. Accumulating studies have showed that heterochromatin regions are replicated at middle and late S phase, but euchromatin regions prone to be replicated at early S phase. In addition, according to Anja Groth lab's results, EdU labeled the vast majority of H3K27me3 peaks (~60-70%) in HeLa S3 cells at middle S phase (Nazaret Reverón-Gómez, et al. Mol Cell, 2018). In our study, we found that the S phase lasts ~4 hrs for mESCs, and cells mainly entered middle S phase after 2 hrs (T0) of releasing (Response1. Fig.1A). These results suggest that this time point is suitable for labeling H3K27me3 regions in mESCs. As expected, genome coverage analysis showed that EdU incorporated into ~70% of H3K27me3 peak regions within 15 mins in the synchronized experiments (Response1. Fig.1B). In order to improve labeling coverage, we performed experiments in asynchronous mESCs with extended EdU-labeling time (20 mins), genome coverage analysis showed that ~80% of H3K27me3 peak regions are labeled by EdU in this condition, which largely overlapped with H3K27me3 peak regions labeled in synchronized mESCs (Response1. Fig.1B and 1C). In addition, ChOR-seq revealed that H3K27me3 exhibited similar restoration kinetics on the nascent chromatin as revealed in synchronized mESCs (Response1. Fig.1D, also in main manuscript Fig.S1G). However, the time window suitable for mapping H3K27me3 restoration is different in synchronized and asynchronous mESCs (0 2 4 6 hrs VS 0 1 2 3 hrs after EdU labeling) (in manuscript Fig.1B and Fig.S1G). In ChOR-seq, to prevent cell entering next S phase, we blocked mESCs with thymidine 3 hrs after EdU labeling (T3) in synchronized experiments, which endow us tracing the restoration of H3K27me3 during one cell cycle. However, in asynchronous condition we found that a portion of late S phase cells rapidly pass through G2/M and next G1 phase, and enter next S phase after 3 hrs of EdU labeling. Addition of thymidine before T3 would cause S phase arrest for mESCs that are in early S phase when we label them with EdU. Hence, thymidine blockage at T3 time point in asynchronous mESCs is not efficient as that in synchronized mESCs. Consistent with this, ChOR-seq showed a reduction of H3K27me3 signals at its peak regions after T3 (Response1. Fig.1E). For these reasons, we represented and analyzed asynchronous ChOR-seq (T0-T3) in the manuscript and got similar results. Anyhow, these results

suggest that synchronized and asynchronous conditions are both suitable for studying epigenetic restoration post-replication, but each condition has its own pros and cons.

Response1. Figure1. (A) Cell cycle process of asynchronous mESCs measured by flow cytometry; (B) Percentage of H3K27me3 peaks labeled by EdU in synchronized and asynchronous mESCs; (C) Venn diagram depicting the overlapping of H3K27me3 regions labeled by EdU in synchronized and asynchronous mESCs; (D) Time series cluster analysis of the restoration pattern of H3K27me3 at H3K27me3-enriched peak regions in asynchronous mESCs; (E) Metaplot analyses of the restoration of H3K27me3 post-replication in asynchronous mESCs.

2. There is little information provided about the data analysis, but it appears that the ChORseq analysis is done without filtering for replicated regions, which is essential in a synchronization set up. By including regions that are not covered by the EdU labelling, conclusions will be based on changes in background signal (i.e, regions that have not been replicated in a high proportion of the cells or not replicated at all in the 15min EdU pulse during early/mid S).

Response: We thank the reviewer for raising this point. In the revised manuscript, we have filtered EdU-labeled H3K27me3 regions from whole H3K27me3-enriched

regions, and revealed that EdU labels almost 70% and 80% of H3K27me3 peak regions in synchronized and asynchronous mESCs, respectively (Response1. Fig. 1B). In this case, we reanalyzed the restoration kinetics of H3K27me3 at EdU-labeled H3K27me3 regions and draw the same conclusion as previously gotten. We have modified figures (Figures 1B and 1D; Figures 2B-2L; Figures 4B and 4C, 4E-4J; Figures S2A-S2E; Figures S4E and S4G; Figures S5A and S5B) and revised the manuscript in page 4). In addition, we have added “Replicated regions define and coverage calculation” in the revised Methods section in page 21.

3.The use of a synchronization setup is also problematic for the comparison between WT and H1 TKO cells, as it assumes that the same set of regions will be replicating during the EdU pulse in early/mid S in both cell lines. This problem would be circumvented by doing ChORseq in asynchronous cells.

Response: We agree with reviewer that it would be a problem to investigate the role of H1-compacted chromatin on H3K27me3 restoration if H1-TKO significantly alters the timing of replication. So, we analyzed the genome coverage of EdU in wild-type and H1-TKO mESCs and found that a large set of genome regions are simultaneously labeled by EdU in synchronized wild type (~69.84% of all labeling regions) and H1-TKO (~70.62% of all labeling regions) mESCs (Response1. Fig.2A left). However, under asynchronous condition, we found that ~52.78% of EdU-labeled genome regions in wild type mESCs are overlapped with 45.88% of EdU-labeled genome regions in H1-TKO mESCs (Response1. Fig.2A right). When analyzing H3K27me3 peaks regions, we found that ~77.83% of EdU-labeled H3K27me3 regions in wild type mESCs are overlapped with 87.65% of EdU-labeled H3K27me3 regions in H1-TKO mESCs (3258 EdU-labeled regions) under synchronized condition, and ~77.22% of EdU-labeled H3K27me3 regions in wild type mESCs are overlapped with ~74.03% of EdU-labeled H3K27me3 regions in H1-TKO mESCs (3725 EdU-labeled regions) under asynchronous condition (Response1. Fig.2B, added in revised manuscript Fig.S4D). These results suggest that H1-TKO indeed partially alters replication timing, while the replication domain used in H3K27me3 peak regions are preserved in wild type and H1-TKO mESCs at a large extent. In addition, we also performed ChOR-seq in HA-AID-H1c/e inducible degradation system, which largely circumvents this concern (Response1. Fig.3, also in manuscript Fig.5). We have modified main text

(page 5, 8-10) and “Methods” (page 17) in the revised manuscript.

Response1. Figure2. (A) Venn diagram depicting the overlapping of EdU-labeled genome regions in wild type and H1-TKO mESCs under synchronized and asynchronous conditions; (B) Venn diagram depicting the overlapping of EdU-labeled H3K27me3 peak regions in wild type and H1-TKO mESCs under synchronized and asynchronous conditions, respectively.

4.The information provided on data analysis is very limited. The authors refer to the software but do not specify how they normalize the signal, how the different units are computed (for example, read density) or how peaks are defined (from ChIP, from ChORseq...). Without this information it is not possible to evaluate the conclusions based on quantitative differences, for example. They do also not specify how the different gene categories are defined; i.e. H3K27me3 peaks are generally not found on housekeeping promoters so what are they looking at?

Response: We thank reviewer for this suggestion. We have provided information about data analysis in the revised manuscript. ChIP-seq and ChOR-seq reads were normalized with fly reference as previous study described. Enriched peaks were called using MACS. The p-value cutoff for peak detection was $1e^{-5}$ with MACS14 (added in Method section). The analyzed regions in the manuscript are all EdU labeled H3K27me3 peaks and regulated genes which with EdU labeled H3K27me3 peaks at promoter regions. We collected the house-keeping, es-specific, bivalent and tissue-specific gene lists from publications (Constance M. Smith et al. Nucleic Acids Res, 2019; Bin li et al. Sci Rep, 2017; Philipp Voigt et al. Genes Dev, 2013; Noriyuki Suzuki et al. J Toxicol Sci, 2011; Alfonso Martinez Arias et al. Curr Opin Cell Biol, 2011; Tüzer Kalkan et al. Development, 2017) and divided H3K27me3 targets genes into 4 categories. Hence, these genes are all with H3K27me3 modification at promoter regions. We have modified the manuscript in page 6. In addition, restoration rate distribution, replicated regions define and coverage calculation are all added in the “Method” session.

5.The major conclusions of Figure 3 and Figure 5 were published previously by Fan et

al., 2005 and Willcockson et al., 2021, reducing overall novelty of the work. H3K27me3 restoration kinetics was also previously shown to be fastest at sites with high K27me3 and PRC2 binding (Reveron-Gomez et al., 2018). It would be advisable to cite these publications when concluding on experiments that validate or corroborate previously findings (even if the papers are cited elsewhere in the manuscript). Given the K27me3 level is 2-fold reduced in the H1 TKO cells, it not clear whether the lower K27me3 or lack of H1 is responsible for the reduced restoration kinetics. An inducible system with similar baseline prior to replication would be preferable. The authors could also compare in WT cells restoration kinetics of regions with similar K27me3 levels and different H1 density. There are conclusions on causality based on correlations mainly concerning the importance of restoration rates for gene regulation.

Response: We agree with reviewer for this concern. To elucidate whether the reduced H3K27me3 level would affect the restoration of H3K27me3 in H1-TKO mESCs, we generated an inducible H1c/e degradation system, in which the HA-AID was knocked into the N-terminus of H1c and H1e genes simultaneously using CRISPR/Cas9 mediated-gene editing in 9×MYC-TIR1 stable expression mESCs (HA-AID-H1c/e cell line). In this system, the HA-AID-H1c/e proteins are gradually degraded through ubiquitin-proteasome system when IAA (indole-3-acetic acid) was added into the culture medium (Response1. Fig.3A and 3B). We found that the synthesis and degradation of HA-AID-H1c/e reach equivalent at ~12 hrs point after IAA treatment, however the H3K27me3 level don't significantly changed before ~24 hrs, endowing us at least ~12 hrs to investigate the restoration kinetics with an equal H3K27me3 initial level. In addition, the results are also consistent with the original findings that reduction of H3K27me3 mainly caused by passive dilution during cell cycle other than turnover and demethylation (Nazaret Reverón-Gómez, et al. Mol Cell, 2018). Basing on these findings, we first cultured HA-AID-H1c/e cell line in IAA containing medium 6 hrs then equally divided the cells into 2 sets (Response1. Fig.3A-C): One set kept being treated with IAA (low HA-AID-H1c/e group), the other set was changed into fresh culture medium without IAA, in which the H1c/e proteins are gradually restored to normal levels during ~6 hrs (HA-AID-H1c/e restoration group). Subsequently, we performed ChOR-seq at 0, 1, 2, 4 time points after IAA withdraw. In this case, ChOR-seq results further support that H1-mediated chromatin compaction plays an important role in promoting H3K27me3 restoration after DNA replication (Response1. Fig.3E-3H). These results are also showed in revised Fig.5 and manuscript in page (9-10).

Response1. Figure3. H1-mediated chromatin compaction facilitates the restoration of H3K27me3 following DNA replication. (A) Experimental flow graph of ChOR-seq in asynchronous HA-AID-H1c/e mESCs. The HA-AID-H1c/e mESCs were treated with IAA for 6 hrs and labeled with EdU 20 mins immediately. Nascent chromatin was harvested immediately after EdU labeling (T0), mature chromatin was collected at different time points (T1, T2, T4) following chasing with IAA (+IAA) or without IAA (-IAA). (B, C) Western blots showing HA-AID-H1c/e, H3K27me3 and H3 levels with or without IAA treatment at different time points. (D) Representative flow cytometry profiles of EdU-labeled cells in nascent samples. (E, F) Average profiles of H3K27me3 ChOR-seq signals at different time points post DNA replication, centered at EdU-labeled gene TSSs in -IAA (E) and +IAA (F) treated mESCs. (G) Snap shot of tracks of H3K27me3 signal at different time points in -IAA and +IAA treated mESCs. (H) Boxplot showing the comparison of restoration rate of H3K27me3 labeled by EdU among -IAA and +IAA mESCs. The P values are calculated according to Wilcoxon signed-rank test.

Frankly, comparing the restoration kinetics of regions with similar H3K27me3 levels and different H1 density is a good idea to elucidate the role of H1-compacted chromatin in regulating the restoration of H3K27me3 post-replication. However, we unfortunately can't collect a set of these regions to perform statistical analysis, because we found that the levels of H3K27me3 are broadly positive correlated with the levels of H1 proteins on chromatin genome-wide (Response1. Fig.4, also in manuscript Fig.3A and 3B).

Response1. Figure4. The levels of H3K27me3 are positive correlated with the levels of H1 on chromatin. (A)Heat map analysis of H3K27me3 and H1e signals across H3K27me3-enriched regions (± 3 kb), ranked from highest to lowest ChIP-seq signals of H3K27me3 in WT mESCs; (B) ChIP-seq cumulative enrichment of H3K27me3 deposition centered at the peak summit of the H3K27me3, the H3K27me3 peak regions were clustered into 5 groups based on the levels of H1e.

In addition, we thank reviewer to point out the issue about the causality between H3K27me3 restoration and gene silencing post-replication. On this topic, we fully agree with reviewer that just basing on our present results, there are no solid evidences to support this conclusion. In the revised manuscript we have toned down the conclusion about this topic in "Results and Discussion" section (page 5 and 12). Besides, we have cited the publications related to Figure 3 and Figure 5 (Figure 3 and Figure 6 in revised manuscript) and linked these findings to the experiment conclusions in revised manuscript (page 6, 8 and 11).

Minor points

1.It is unclear what data was produced in this manuscript and what data comes from published datasets. There are no accession numbers or references to any published data.

Response: We thank reviewer for this suggestion. We have provided the information in

Data Accession of revised manuscript and in “reporting summary” accompanying with this manuscript.

2.The authors refer to different genomic regions without defining them or providing any information on their overlap, e.g. they classify first restoration kinetics of peaks and then calculate the restoration kinetics to pre-defined promoters (ES, housekeeping, bivalent etc.). It is also not clear if these promoters have H3K27me3.

Response: We are sorry for the confusion. Indeed, in our study all genes that we analyzed are H3K27me3 targets. To achieve this, we first collected the gene list of various categories (house-keeping, es-specific, bivalent and tissue-specific) from previous publications, and then filtered EdU-labeled H3K27me3 target genes from individual gene list. By this means, we determined the four categories of H3K27me3 target genes for further analysis. It is worth noting that the number of bivalent genes accounts for the majority of H3K27me3 target genes (Response1. Chart 1). We have added the details to the revised manuscript (page 6).

	All H3K27me3 target genes	EdU-labeled genes (synchronized)	EdU-labeled genes (asynchronous)
Bivalent	2261	1378	1488
Tissue specific	1356	811	881
House-keeping	338	202	200
ES	62	39	49

Response1. Chart 1. The gene numbers of all H3K27me3 targets in four functional gene clusters.

3.Details missing in the figures and figure legends, mostly units and data source. Details missing in the methods, e.g there is a second thymidine addition in the schematic in Fig.1 but this is not clear from the methods.

Response: We thank reviewer for this suggestion. We have revised the manuscript thoroughly and added the missing details in Figure 1A, figure legends (1A, 1E, 2D-2G, 2J, 4I) and Methods sections (page 17) that were highlighted in the manuscript. We also added data source in Data Accession section.

Reviewer #2 (Response to the reviewer):

Mechanisms of propagation and maintenance of epigenetic regulation across cell cycle are of significant interest, both because of fundamental importance to developmental biology, and due to emerging understanding that many human disorders are driven by misregulation of gene expression. The idea that repressive chromatin modifications are particularly dependent on the “inheritance” mechanisms during S phase has been developed in significant detail by many laboratories, including Reinberg (e.g., Reinberg and Vales, *Science* 2018, and Escobar et al., *Cell* 2019), Grewal (Holla et al., *Cell* 2020), Crabtree (Chory et al., *Mol Cell* 2019) and others. Likewise, emerging interest in the function of linker histone H1 is highlighted by recent publications from Skoultschi and Melnick laboratories (Willkockson et al., *Nature* 2021 and Yusufova et al., *Nature* 2021) demonstrating, among others, the effect of H1 loss on core histone modification landscape. While previous results imply that PRC2 methyltransferase complex is stimulated by H1 incorporation in vitro (Martin et al., *JBC* 2006), mechanistic in vivo evidence is incomplete. Skoultschi and Fan groups have previously demonstrated that loss of H1 in mES cells impairs differentiation [e.g., Zhanmg et al., *Plos Genet* 2012] but given the complexity of the phenotypes, specific relationship between H1 incorporation and PRC2-driven H3K27 methylation must be studied further. The current study attempts to tackle this question using several experimental approaches. First, the authors study H3K27me dynamics in post-replication chromatin using ChOR-Seq. Second, they generate and characterize H1 triple-knockout cell line. Third, they perform several biochemical assays to assess the impact of H1 incorporation on PRC2 function in vitro. While the paper is undoubtedly timely and the question under investigation is of broad interest and significance, I have several concerns about both the experimental setup and outcomes.

1. I have some concern about the ChOR-Seq setup. Replication is not uniform across the S-phase - with repressive chromatin generally contained in late-replicating domains. If the cells are indeed synchronized at the onset of S phase (as stated at line 187 of supplementary material - although Fig S4A argues that quite a few cells are in the middle of S?), and S phase takes several hours to complete, I would expect that early-replicating domains, predominantly found in active chromatin, would be the first to incorporate EdU - and more delayed-replicating domains, located in K27mrepressed (and ultimately K9me-repressed) chromatin would engage in replication much later. A

recent paper by Glibert group (Zhao, Sasaki and Gilbert, Genome Biol 2020) has mapped several classes of initiation zones in mES cells, with roughly 1500 initiating early, and ~1000 initiating later in the S-phase. At the very least, I'd like to know how do these correspond to ChOR-Seq data in the current manuscript? Given that, to the naked eye, most robust changes in K27me3 happen between T0 and T2(Figure S1B, and Figure 4A, B, D), I'd like to know more about these regions (it is somewhat counterintuitive that K27me3-rich regions are replicated within first two hours of release).

Response: We are sorry for the confusion about ChOR-Seq setup. In this study, we first synchronized the cells at the G1/S phase boundary using 2 mM thymidine (single block) and then released synchronized mESCs into cell cycle by changing fresh medium. After 2 hrs of releasing (T0), we labeled replicated DNA with 20 μ M EdU 15 mins when most of cells are in middle/late S phase (main manuscript Fig.1A and Response2. Fig.1A). Consistent with this, genome coverage analysis showed that EdU incorporated into ~70% of H3K27me3-enriched regions (Response2. Fig.1B). It argues that heterochromatin is replicated later than active chromatin. Besides, our ChOR-Seq and enrichment analysis showed that a portion of H3K27me3-highly-enriched peaks (28.73%) are restored rapidly during T0-T2 (main manuscript Fig.1B and Fig.S1F), demonstrating H3K27me3 is rapidly restored at dense heterochromatin regions following DNA replication (main manuscript Fig.1E). In addition, T0-T2 is the first 2 hrs of chasing (2.15-4.15 hrs after release).

Response2. Figure1. (A) Cell cycle process of asynchronous mESCs measured by flow cytometry; (B) Percentage of H3K27me3 peaks labeled by EdU in synchronized and asynchronous mESCs; (C) Venn diagram depicting the overlapping of H3K27me3 regions labeled by EdU in synchronized and asynchronous mESCs; (D) Time series cluster analysis of the restoration pattern of H3K27me3 at H3K27me3-enriched peak regions in asynchronous mESCs; (E) Metaplot analyses of the restoration of H3K27me3 post-replication in asynchronous mESCs.

2. Perhaps even more puzzling, looking at the data in Fig S1B, I DO NOT see three distinct classes of K27me3 restoration. Even though the peaks are stated to be scaled by parental K27me3 density (or peak height?), they all look virtually identical both in their dynamics (change from T0 to T6) and original peak intensity (top to bottom in “Parental” heatmap. Given that much of the paper is based on interpretation of these data, it’s critical to show that there are indeed distinct restoration clusters. Figure 1B looks like it, and perhaps some of the discrepancy is due to the scale differences (S1B is scaled -3 to 3, and 1B is scaled -1 to 1). Yet I do not see how clusters A, B and C are derived from raw data shown in S1B.

Response: We thank reviewer for this suggestion. Accordingly, we have shown the

clustered H3K27me3 signals by a new Heat-map in revised manuscript (in manuscript Fig.S1E and Response Fig.5). Figure 1B and Fig S1E are different presentation of the same clustered ChOR-seq data.

Response2. Figure2. Heatmap analysis showing the restoration clusters of H3K27me3-enrichment regions genome wide.

3. As overall steady-state levels of K27me3 are different between the two cell lines used in the study (see fig. 3C, D - and previously reported e.g. by Willcockson et al., Nature 2021), and pre-existing K27me3 levels determine the rate of K27me3 spreading, I am not convinced that this system allows the authors to unequivocally state that H1 loss is directly affecting the rate of spreading. We know that H1 appears to stimulate PRC2 even in di-nucleosomal substrates (first reported by Martin et al., JBC 2006) - I think the jump to compaction as mechanism is intriguing and very plausible, but is not fully supported by data. Further, comparing data in 2C-G, I am left with the impression that the greatest predictor of K27me3 restoration rate (Fig. 2C) is K27me3 read density (Fig 2D) - not the H1e read density (Fig 2E).

Response: We agree with reviewer for this concern. To circumvent this problem, we generated a H1c/e inducible degradation system (HA-AID-H1c/e cell lines). In this system, the HA-AID-H1c/e proteins are gradually degraded through ubiquitin-proteasome system after adding IAA (indole-3-acetic acid), but bulk H3K27me3 do not significant changed during the tracking period of H3K27me3 restoration. Then we performed ChOR-seq and found that H1c/e reduction resulted a significant delay of H3K27me3 restoration after DNA replication (Response2. Fig.3A-H, revised

manuscript Fig.5), which supports the conclusion that H1-mediated chromatin compaction have an important role in promoting H3K27me3 restoration after DNA replication.

Response2. Figure3. H1-mediated chromatin compaction facilitates the restoration of H3K27me3 following DNA replication. (A) Experimental flow graph of ChOR-seq in asynchronous HA-AID-H1c/e mESCs. The HA-AID-H1c/e mESCs were treated with IAA for 6 hrs and labeled with EdU 20 mins immediately. Nascent chromatin was harvested immediately after EdU labeling (T0), mature chromatin was collected at different time points (T1, T2, T4) following chasing with IAA (+IAA) or without IAA (-IAA). (B, C) Western blots showing HA-AID-H1c/e, H3K27me3 and H3 levels with or without IAA treatment at different time points. (D) Representative flow cytometry

profiles of EdU-labeled cells in nascent samples. (E, F) Average profiles of H3K27me3 ChOR-seq signals at different time points post DNA replication, centered at EdU-labeled gene TSSs in -IAA (E) and +IAA (F) treated mESCs. (G) Snap shot of tracks of H3K27me3 signal at different time points in -IAA and +IAA treated mESCs. (H) Boxplot showing the comparison of restoration rate of H3K27me3 labeled by EdU among -IAA and +IAA mESCs. The P values are calculated according to Wilcoxon signed-rank test. (I) The activity of PRC2 on poly-nucleosome and H1-compacted chromatin substrates. (J) The activity of PRC2 on mono-nucleosome absent or present of H1 binding. The signals of H3K27me3 products (I, J) were detected by ³H-methyl autoradiography, and the amount of substrate was detected by coomassie staining.

In addition, using in vitro HMT assays we found that PRC2 and H3K27me3 indeed have a positive feedback loop (consistent with previous findings), and H1 can further enhance PRC2 HMT activity on poly-nucleosomes through compacting chromatin fibers (Response2. Fig.3I and J, revised manuscript Fig.6 and Fig. S6), suggesting that “positive feedback loop” and H1-compacted chromatin are two independent mechanisms, and they could cooperatively promote the robust spreading of H3K27me3 and the fast restoration of H3K27me3 post-replication in cells. We have shown the data related to H1c/e inducible degradation system in “Fig.5” in revised manuscript. We also described this finding in the “response letter” to reviewer 1.

4. The RNA-Seq in H1 TKO cells is a bit puzzling given almost identical number of genes going up and down. I would have expected more genes being upregulated given how dramatically H3K27me3 appears to be affected (and given what we know about H1 as a repressor). The GSEA plots are not informative as shown. Do the upregulated genes encompass sets found within cluster A? What are the downregulated genes?

Response: The well-known function of H1 is a transcriptional repressor, since it has been revealed H1s can inhibit the binding of transcription factors and pol II onto enhancer and promoter DNA elements through locking entry/exit sites of DNA and compacting chromatin fibers. However, gene expression analysis after knock-down and Knock-out of H1s have shown that the reduction of H1 protein levels did not cause global up-regulation of transcription but rather mis-regulation a specific set of genes including both up-regulation and down-regulation in tetrahymena thermophile, chicken

and mouse (Hergeth SP and Schneider R. EMBO Rep, 2015). Consistent with this, a mount of studies have demonstrated that H1 not only block the binding of proteins (including activators and repressors) to chromatin but also serve as a recruitment platform for both activators and repressors (e.g. activator including Cul4A PAF1, repressors including Su(var)39 HP1 and L3MBTL3) (Hergeth SP and Schneider R. EMBO Rep, 2015). In addition, it has been shown that the binding of H1 to the MMTV promoter induces a distinct chromatin conformation that facilitates the binding of hormone receptor and transcription factors and leads to efficient transcriptional activation (Hergeth SP and Schneider R. EMBO Rep, 2015). Besides, Yuhong Fan's lab previously found that single-H1 KO and H1-TKO in mESCs mainly causes down-regulation of Hox genes other than up-regulation (Yunzhe Zhang, et al. PloS One, 2012), and several groups suggested that H1 could mis-regulate gene expression through altering DNA methylation and high order chromatin structure (Michael A Willcockson, et al. Nature, 2021; Geert Geeven, et al. Genome Biol, 2015). Hence, the alteration of mRNA in H1-TKO mESCs should be a superimposition effect of these mechanisms mentioned above. However, in fact there are studies showing H1 knockout mainly cause up-regulation of genes in somatic cells, such as lymphocytes (Nevin Yusufova, et al. Nature, 2021), suggesting that H1s differentially regulate gene expression among distinct type of cells.

In our revised manuscript, we filtered mis-regulated H3K27me3 targeting genes (299) and found that 56.19% (168) were down-regulated and 43.81% (131) were derepressed among mis-regulated H3K27me3 targeting genes. These results indicated that H1 also regulates other transcriptional regulation pathways at H3K27me3-enriched regions such as DNA methylation, H3K9me and HP1 and so forth, and it seems that the pathway we identified maybe dominantly involved in regulation of these up-regulated genes (131) compared with other mechanisms, at least, in mESCs. Distribution analysis showed that cluster A covered a large set of these mis-regulated including up- and down regulated genes (Response2. Fig.4), indicating that the chromatin states in cluster A should experience a dramatical change following H1c/d/e knockout. We have revised the manuscript in page 8.

In addition, we deleted the GSEA plots in the revised manuscript, because we agree that there is no conception novel for the present study.

Response2. Figure4. The percentage distribution of mis-regulated genes among different H3K27me3 restoration clusters including super-fast (A), fast (B), slow (C), bottom slow (D).

5. The previous concern could be partially alleviated by a more robust distinction between recruitment and spreading. If H1 is indeed affecting the spreading dynamics, then the K27me3 peak height should remain roughly similar - but the breadth is expected to be reduced dramatically. At the very least, I'd expect a faster dropoff (such that K27me3 peaks in WT would essentially represent "obtuse angles", and in TKO would be more akin to "acute angles") from the peak center to the edges. The authors do the right experiment in Fig 5E-F, but looking at the data, I don't see a difference in K27me3 peak shape - only an overall decrease in H1 TKO background - therefore, I still can't parse out whether H1 loss causes defect in "recruitment" or "spreading" in vivo.

Response: In targeting experiment (Fig 5E-5F), the shape of H3K27me3 peak is largely dependent on the primer sets used in ChIP-qPCR, which may alter when choosing other sets of primers. To rule out this possibility, we repeated the targeting experiments and quantitated H3K27me3 enrichment by ChIP-seq. In addition, the HMT and spreading activation of PRC2 requires heterodimer of EZH2 and EED subunits, so we this time co-overexpressed HA-EZH2-TetR/FLAG-EED or HA-EED-TetR/FLAG-EZH2 in 8×TetO targeting cells, which could additionally augment the difference of H3K27me3 spreading between wild type and H1-TKO mESCs compared with single HA- EZH2-TetR overexpression. Our ChIP-seq results showed that targeting HA-PRC2-TetR efficiently deposits H3K27me3 across 8×TetO sites with width distribution in wild type cells, and the peak height of H3K27me3 is 10× lower and the width is narrowed in H1-TKO (vertical ordinate for H1-TKO is zoomed in 3 times in the picture show) (Response2. Fig.5D). In our previous study (Jicheng Zhao et al, NCB, 2020, Fig.3j),

we have tested the effect of H1 loss on the “recruitment/targeting” of HA-TetR-proteins and found that the height of HA-TetR-RING1B peaks in H1-TKO is the same as that in wild type mESCs at 8xTetO peak center (Response2. Fig.5A), suggesting that H1 loss don't significantly intervene the binding of HA-TetR-proteins with TetO arrays.

Response2 Figure5. (A) the binding of RING1B-TetR-HA around 8xTetO sites following overexpression, we also presented this data in our previous publication (Jicheng Zhao et al, NCB, 2020, Figure 3j); (B) the activity of PRC2 on mono-nucleosome with or without H1 binding; (C) the activity of PRC2 on poly-nucleosome and H1-compacted chromatin substrates. The signals of H3K27me3 products were detected by ³H-methyl autoradiography. (D) Genome browser tracks of H3K27me3 peaks across the 8xTetO-integrated locus in WT and H1-TKO mESCs following HA-EED-TetR + FLAG-EZH2 or HA-EZH2-TetR + FLAG-EED.

In addition, we found that H1 didn't affect PRC2 activity on mono-nucleosome

(manuscript Fig.S6E, Response2. Fig.5B), demonstrating that H1 don't intervene the interaction between enzymes and nucleosomes. Besides, it is widely accepted the notion that open chromatin would be more accessible for enzymes than H1-compacted chromatin, so PRC2 should inert on H1-condensed chromatin if no other mechanism exists, but we in fact observed the opposite results (PRC2 is more active on H1-condensed chromatin) (Response2. Fig.5B and 5C; also showed in main manuscript Fig.6B and Fig.S6E). Therefore, these results altogether suggest that H1-compacted chromatin has an important role in H3K27me3 restoration by promoting the spreading of H3K27me3. According with our results, it seems that the recruitment, both positive feed-back loop of PRC2-H3K27me3 and H1-compacted chromatin contribute to the height of H3K27me3 signals immediate around 8×TetO, which should be a cooperative effect of these three mechanisms at least.

6. Several essential controls are missing:

a. While the original H1 TKO lines generated by Skoultchi lab have been extensively characterized and show about 2-fold reduction of total H1 levels, the newly generated lines reported in the paper require additional characterization. What is the genomic landscape of CRISPR-generated lines? Did Cas9 editing create large deletions or small substitutions? How were the clones isolated (or are these non-clonal lines?) What are the levels of H1 (can be estimated by HPLC, or even by Coomassie staining, since H1 isoforms have an obvious migration pattern - this is critical since H1 genes are dosage compensated!)

Response: In order to generate H1-TKO mESCs, we chose a guide RNA that simultaneously targets H1c/d/e genes, and ligated it into pX260-Cas9 vectors, the guide RNA sequence was showed in “methods section”. After transfection, we plated cells into 3 10 cm culture plates with serial diluted density and screened with 1ug/ml puromycin. 10 days later, single cell clones were picked out and the genotypes (frameshift mutations) were identified by PCR-coupled DNA sequencing (Response2. Fig.6A). To evaluate the reduction of H1 in H1-TKO mESCs, we performed acid extract for histones from wild type and H1-TKO mESCs and evaluated the reduction of H1 in H1-TKO mESCs using Coomassie staining methods. To achieve this, we linearly (100% to 50%) loaded the extracted histone samples that were quantified by BCA during SDS-

PAGE gel running (Response2. Fig.6B top). After staining, we respectively estimated the “gray value” of linker histone H1s and core histone for each lane using ImageJ (subtracting the background). Then we calculated the ratio of H1/core histones for wild type and H1-TKO at the series of loading condition. Next, the reduction of H1s in H1-TKO mESCs relative wild type mESCs were estimated by the formula showed in Response2. Fig.6B (bottom) under the premise that the wild type and H1-TKO samples have the similar “gray value” for core histones, such as the pairs between VS1 or the pairs between VS2. This result confirmed that H1-TKO causes ~2-fold reduction of bulk linker H1 levels after knockout of H1c/d/e as described by Skoultchi lab previously.

To further verify that H1.2 (H1c)/H1.3 (H1d)/H1.4 (H1e) proteins are absent in our H1-TKO cell line, we analyzed the extracted histones from wild-type and H1-TKO cells by mass spectrometry (Response2. Fig.6C and 6D). The result showed no peptide uniquely identified from histone H1.2/H1.3/H1.4, indicative of a successful knock-out of all three isoforms, albeit the remaining two isoforms H1.1 and H1.5 were both identified in high abundance (C: number of PSMs in WT and H1-TKO samples identified by MS). A close analysis on all unique peptide identified by mass spectrometry further indicated that only H1.1 and H1.5 could be identified in the H1-TKO cell line, confirming the cell identity (D: number of PSMs for each unique peptide from all five H1 isoforms identified by MS).

Response2. Figure6. Identification of H1-TKO mESCs generated in our lab. (A) DNA sequencing results of H1.2, H1.3 and H1.4 coding regions that were amplified by PCR from H1-TKO mESCs; (B) Coomassie blue staining of acid extracted histones for wild type and H1-TKO mESCs (top), a chart showing the ratio of H1/core histones for wild type and H1-TKO mESCs and the formula for calculating the reduction of H1 proteins in H1-TKO mESCs (bottom). (C) Number of PSMs in WT and H1-TKO samples identified by MS. PSM: Peptide Spectrum Match; (D) Number of PSMs for each unique peptide from all five H1 isoforms identified by MS.

b. are K27me3 levels uniform in these cells? (metastable mES cells grown in serum/LIF media have significant variability in H3 K27me3 levels - are the K27me3 “low” cells cycling faster or slower? (basically, which cell state contributes most signal to the ChOR?). I am not sure “original” TKO cells from Skoultschi lab were ever tested for K27me3 heterogeneity, and interestingly, the original report (Fan et al., Cell 2005) did not report significant changes in K27me - but it seems like an important question. Perhaps analyses similar to Fig S4A can be done to assess the K27me3 levels during the cell cycle?

Response: In order to clarify if metastable mES cells grown in serum/LIF media have significant variability for H3K27me3 levels, we have performed FACS for wild type and H1-TKO mESCs using double staining by H3K27me3 antibody and PI. Interestingly, the results showed that the H3K27me3 is relatively homogenous in these cells (Response2. Fig.7A-7C), which rules out the possibility raised by review. In addition, we believe this concern can also be resolved by the H1 inducible degradation

experiments. As mentioned, we also reanalyzed the studies carried out by Skoultchi lab and confirmed that they also found ~2 fold reduction of H3K27me3 in H1-TKO mESCs (Response2. Fig.8). Besides, in the previous studies by Skoultchi lab they did the immunofluorescence for H3K27me3, but didn't show the results in their paper.

Response2. Figure7. Flow cytometry profiles of H3K27me3 intensities at single cell level, cell cycle was evaluated by DNA content with propidium iodide (PI) staining. H3K27me3 was sequentially stained by anti-H3K27me3 and ant-rabbit IgG (FITC). (A) scattergram of H3K27me3 and PI intensity in the negative control, K3K27me3 only, PI only, wild type R1 and H1-TKO mESCs. (B) Histogram analyses of H3K27me3 intensity in the negative control, R1 WT and H1-TKO cells. (C) Merged expression profiles H3K27me3 in R1 WT and H1-TKO cells.

[redacted]

Response2. Figure8. Western blots for H3K27me3 and other histones marks in WT and H1-TKO mESCs (Fan, et al. 2005, Cell)

c. To add to Fig S4A, I'd like to see more detailed quantification of cell cycle progression in these cells. E.g., it appears that T4 plots are a bit different - is it an outlier or part of the phenotype?

Response: Thank reviewer for pointing this issue out. We reanalyzed the biological repeated experiments data for cell cycle analysis and confirmed that the cell cycle of wild type and H1-TKO are roughly the same (Response2. Fig.9). The cell cycle difference between wild type and H1-TKO at T4 is a deviation in each batch of FACS. Basing on our experiences, the cell number and time spent on ice during staining before FACs all slightly contributes to the signal deviation. In the revised manuscript, we replaced old data by biological repeated data and we also added the vertical ordinate in Fig S4A.

Response2. Figure9. Cell cycle progression was monitored by FACS analysis of DNA content in wild type and H1-TKO mESCs.

d. In the biochemical experiments reported, the authors need to show that H1e expressed and purified is full length. H1s are notoriously hard to purify without degradation, and simple His-tag system may be insufficient (several workarounds have been reported in the literature, including dual tags - Osunsade et al., Biochemistry 2019). Coomassie gels shown in Fig. 5 should be cropped above 30 kDa size to account for H1.

Response: Thank reviewer for pointing this issue out. In our study, in order to generate enough highly purified recombinant H1e proteins we have optimized the process for H1 purification. Briefly, we first harshly washed bacteria inclusion bodies, which contain recombinant H1e proteins, to remove contaminants as possible as we can.

Second, we resolved H1e from bacteria inclusion using 1M NaCl, then sequentially purified with Heparin column and hydroxyapatite column as we described in the methods section. Finally, we identified the purity and stability of H1e using SDS-PAGE and Coomassie staining, and we found that the recombinant H1e purified using this protocol is pure and relative stable (vast majority of the H1e is full length) even after a long period of storage at -80°C (Response2. Fig.10). We have shown the Coomassie staining result of H1e in the revised manuscript (Figure 6 and Figure S6).

Response2. Figure10. Coomassie blue staining the purified full length and truncated linker histone H1E in SDS-Page gel.

7. Throughout figures (e.g., fig 1B, E, fig 3 A, D - and several supplementals) labels in heat map color scales are missing. Whether these are z-scores, log₂ fold change, log₂ of read counts, or other value - should be labeled as such. There is no question that the paper is timely and of broad interest, yet given that in vivo experiments shown have several significant caveats, and in vitro experiments largely replicate known results (Martin et al. 2006 and Willcockson et al., 2021), I am not fully convinced the paper is ready for publication until the authors are able to disentangle defect in K27me₃ spreading from overall decrease in K27me₃ observed in their system at steady state.

Response: Thank reviewer for these suggestions. Accordingly, we have added all necessary information in the revised manuscript (Figures 1B, 1E, S1G and S4E; Figure legend of Figures S1E, Figures 3A and 3D).

Reviewer #3 (response to the reviewer):

In the manuscript “Histone H1 facilitates restoration of H3K27me3 during DNA replication by chromatin compaction”, Liu et al. reported that H1 facilitates the rapid post-replication restoration of H3K27me3 on repressed genes using ChOR-seq. The authors also showed that H1 facilitates the propagation of H3K27me3 by PRC2 in vitro and that the H3K27me3 restoration on nascent DNA is compromised in H1-TKO mESCs. Overall, the experiments were well performed, and results were clearly presented. ChOR-seq provided specifics on the rapid restoration of H3K27me3 following DNA replication, although H1 facilitating H3K27me3 propagation is somewhat expected given the substrate preference of PRC2 for H1 containing nucleosomes (Martin C 2006). The drastic effects by H1-TKO in mESCs observed in this study (dramatic gene expression changes (>1200 genes) and H3K27me3 ChIPseq), however, are in stark contrast to findings from previous studies. It has been well documented that H1-TKO produces limited expression changes /negligible changes in H3K27me3 ChIP-seq in undifferentiated mESCs but induces dramatic changes in more specialized/differentiated cells (Fan 2005, Zhang 2012, Geeven 2015, Encode, Willcockson 2021, Yusufova 2021, etc.). How many mESC lines were analyzed and how were these H1-TKO mESCs by CRISPR/Cas9 characterized? It’s not clear why the authors used H1-TKO mESCs by CRISPR/Cas9 when only found H1e enrichment with H3K27me3 and in cluster A in mESCs. The significance of H1e in rapid restoration of H3K27me3 in vivo remains to be addressed.

Response: Thank reviewer very much for his/her positive comments on our study. For the effect of H-TKO on gene expression in mESCs, it has been measured using different methods previously. In publication (Fan et al. 2003, Cell), authors compared the gene-expression profile of 6,842 genes by hybridization of biotin-labeled antisense RNA (cRNA) from triple-H1 null and wild-type littermate ES cells to Affymetrix U74Av2 microarrays, and found that 38~ genes were mis-regulated in H1-TKO mESCs. In publication (Zhang et al. 2012, Plos Genetics), author measured <100 genes including pluripotency genes and important developmental genes by PCR SuperArray analysis, showing that majority of these genes showed dramatic change between wild type and H1-TKO mESCs (Response3, Fig.1). In publication (Geert Geeven et al. 2015, Genome Biol), authors used RNA-seq to exam the change of mRNA in H1-TKO mESCs (3 replicates) compared with wild type mESCs (2 replicates) and found that ~600 genes

are mis-regulated in H1-TKO mESCs among non-redundant collection of 20,876 known RefSeq transcripts (Response3. Fig.2A). In our study, we carried out triplicate mRNA-seq both in wild type and H1-TKO (Response3. Fig.2B) that should improve the detection rate for mis-regulated genes, and we use ~30000 genes annotated by “gene code” to filter the mis-regulated genes. Besides, the resolution of mRNA-seq is much higher than microarray used by Fan et al. Hence, we think that it’s reasonable to identify more mis-regulated genes in our studies.

[redacted]

Response3. Fig1. Gene expression profiles in wild type and H1-TKO mESCs and EBs by quantitative PCR SuperArray (left), the significance of the change of genes levels (right). (Zhang et al. 2012, Plos Genetics, Fig.2C and 2D)

[redacted]

Response3. Fig2. Gene expression profiles in wild type and H1-TKO mESCs measured by Mrna-seq. (A) data from (Geert Geeven, et al. 2015, Genome Biol, Fig4.B), (B) data generated in our lab.

According to “why we used H1-TKO mESCs by CRISPR/Cas9 when only found H1e enrichment with H3K27me3 and in cluster A in mESCs”, frankly the idea is inspired by previous other labs and our investigations including (Fan, et al. 2005, Cell; Encode, Willcockson, et al. 2021, Nature; Yusufova, et al. 2021, Nature; Zhao et al. 2020, NCB). In these studies, Western blot and ChIP-seq results indicated that H3K27me3 significantly decreased (Response3. Fig.3 and Fig. 4), but the mechanism is still unknown.

[redacted]

Response3. Figure3. Western blots for H3K27me3 and other histones marks in WT and H1-TKO mESCs (Fan, et al. 2005, Cell)

Response3. Figure4. Western blots for H3K27me3 and other histones marks in WT and H1-TKO mESCs (Zhao, et al. 2020, NCB, Fig.3G)

We agree reviewers that the significance of H1e in rapid restoration of H3K27me3 *in vivo* remains to be addressed. However, according to our study we found that the restoration of H3K27me3 is promoted by H1-mediated chromatin compaction, we believe other H1 isoforms also can promote the restoration of H3K27me3 post-replication although the present ChIP-seq data don't show an obvious enrichment of HA-H1c/d at cluster A genome regions.

REVIEWER COMMENTS

Reviewer #1 (Remarks to the Author):

The authors have addressed several of my concerns and added new data that have strengthened the manuscript substantially. However, there are a few important points that remains to be addressed to make the conclusions robust especially with respect to the new data.

1.

Figure 2J. Add significance to the representation of the different gene categories. E.g. represent odds ratio in a heat map.

2.

Figure 2K/L. Add an extra panel to illustrate H3K27me3 levels on the different gene groups. This will likely illustrate that the H3K27me3 levels on these genes are follow the general principle that higher H3K27me3 levels restore faster.

3.

The T0 timepoint seems to have quite low enrichment. This might not affect the relative restoration kinetics, but at least they should comment on this. Could it reflect different levels of EdU incorporation?

4.

The ChOR-seq analysis should be performed in two replicates as a minimum – all the newly added experiments appears to represent only one experiment.

5.

The new H1-degron experiment in Figure 5 substantially strengthen the authors conclusions regarding the importance of H1 in H3K27m3 restoration. However, the setup is surprising. Why do the authors choose to compare samples that have all been auxin treated for 6hrs and then split in two for continued auxin treatment and auxin removal? The logic would be to compare restoration in the presence or absence of H1 (thus 12 hrs +/- auxin as the T0 timepoint).

It complicates the interpretation that Panel B/C are not quantitative (unlike Fig. 3C). Could the authors provide semi-quantitative westerns to allow proper evaluation of H3K27me3 and H1 levels (or quantitative ChIP?)?.

There is a clear effect in their setup at 4 hrs restoration but very limited effect at T1 and T2. Might this be because there is limited role of H1 early on or because there is not sufficient difference in H1 levels between the two conditions they compare?

If only the 4hr timepoint is different, it might not be appropriate to calculate restoration rates – it should be sufficient show that restoration at T4 is impaired. It would be informative to show restoration of H3K27me3 peaks stratified according to H1 levels (in the format of 5E/F)

The stats provided in H-I are not reflecting biological or technical variability as the data are from one replicate. They should at least use the average of two replicates or show two replicates to demonstrate reproducibility.

6.

One comment in the first revision was that the manuscript generally had a problem with not referencing published work when the authors report similar findings. This has been improved, except in a few places:

a.

It is still unclear what is novel in figure 6 compared to published work. The authors present their data and conclusions in a long paragraph and then at the end state: 'Taken together, our results validated previous findings^{23,37}'. The studies should be cited at the appropriate places, and it should be highlighting what is novel – alternatively the authors should consider to move the data to the supplements if they mainly confirm published literature. Please also include a reference to Willcockson et al., 2021 Nature (Extended data fig. 7).

b.

The authors removed most claims of causality with respect to restoration rate and gene expression, but should correct this in the discussion: 'Importantly, we also found that the restoration rate of H3K27me3 is negatively correlated with the mRNA levels of the genes among tissue-specific, bivalent, ES and house-keeping gene clusters, which ensures the proper gene expression program in mESCs.'

And title on supplementary figure 2: 'Supplementary figure 2. The restoration rate of H3K27me3 is pivotal for the post-replication gene silencing.'

Reviewer #2 (Remarks to the Author):

I am satisfied with the additional experiments and analyses provided in the revised manuscript. I do not have further concerns.

REVIEWER COMMENTS

Reviewer #1 (Remarks to the Author):

The authors have addressed several of my concerns and added new data that have strengthened the manuscript substantially. However, there are a few important points that remains to be addressed to make the conclusions robust especially with respect to the new data.

1. Figure 2J. Add significance to the representation of the different gene categories. E.g. represent odds ratio in a heat map.

Response: We thank the referee for the suggestion. We have added the odds ratio for different gene categories and included the result in the revised manuscript as Fig.S2F (Response Figure.1).

Response Figure.1 (see also in Fig.S2F)

2. Figure 2K/L. Add an extra panel to illustrate H3K27me3 levels on the different gene groups. This will likely illustrate that the H3K27me3 levels on these genes are follow the general principle that higher H3K27me3 levels restore faster.

Response: We thank the referee for the suggestion. We have analyzed the H3K27me3 levels of different gene groups and showed the result in the revised manuscript as Fig.2K (Response Figure.2). The result supports our conclusion that higher H3K27me3 levels restore faster. We also rephased the manuscript as “Consistent with these observations, the restoration rate of H3K27me3 is the lowest for house-keeping genes which have the lowest H3K27me3 level and the highest mRNA levels in mESCs (Figures 2K - 2M)” on Page 6-7.

Response Figure.2 (see also in Fig.2K)

3. The T0 timepoint seems to have quite low enrichment. This might not affect the relative

restoration kinetics, but at least they should comment on this. Could it reflect different levels of EdU incorporation?

Response: The referee has raised an important point here. To address this question, we analyzed the levels of H3K27me3 labeled by EdU across all time points (T0, T2, T4 and T6) and found similar EdU incorporation levels (Response Figure.3). This result indicates that the low enrichment at T0 is not due to insufficient EdU incorporation at the time point, which does not affect the relative restoration kinetics. Thus, we also rephased the manuscript as “Due to DNA replication-dependent dilution of H3K27me3, the T0 timepoint seems to have quite low enrichment. But this might not affect the relative restoration kinetics” on Page 13.

Response Figure.3

4. The ChOR-seq analysis should be performed in two replicates as a minimum – all the newly added experiments appears to represent only one experiment.

Response: We thank the referee for pointing out this issue. We have replicated the ChOR-seq experiments both in asynchronous mESCs and HA-AID-H1c/e cell lines, and analyzed the data accordingly. The analyses showed that the results we showed in previous manuscript are repeatable (Response Figure.4). The newly generated sequencing data have been deposited in NCBI's Gene Expression Omnibus under accession code GSE192984.

Response Figure.4

5. The new H1-degron experiment in Figure 5 substantially strengthen the authors conclusions regarding the importance of H1 in H3K27m3 restoration. However, the setup is surprising. Why do the authors choose to compare samples that have all been auxin treated for 6hrs and then split in two for continued auxin treatment and auxin removal? The logic would be to compare restoration in the presence or absence of H1 (thus 12 hrs +/- auxin as the T0 timepoint).

It complicates the interpretation that Panel B/C are not quantitative (unlike Fig. 3C). Could the authors provide semi-quantitative westerns to allow proper evaluation of H3K27me3 and H1 levels (or quantitative ChIP?)?.

There is a clear effect in their setup at 4 hrs restoration but very limited effect at T1 and T2. Might this be because there is limited role of H1 early on or because there is not sufficient difference in H1 levels between the two conditions they compare?

If only the 4hr timepoint is different, it might not be appropriate to calculate restoration rates – it should be sufficient show that restoration at T4 is impaired. It would be informative to show restoration of H3K27me3 peaks stratified according to H1 levels (in the format of 5E/F)

The stats provided in H-I are not reflecting biological or technical variability as the data are from one replicate. They should at least use the average of two replicates or show two replicates to demonstrate reproducibility.

Response: We thank the referee for the comments.

- (1) In the first revision, the referee concerned about whether the enrichment of H3K27me3 is more important for the restoration of H3K27me3 post-replication than H1-mediated chromatin compaction, because there is a positive feedback loop in the deposition of H3K27me3 between PRC2 and existing H3K27me3. Since the steady knockout of H1c/d/e (H1-TKO) results in a remarkable reduction of H3K27me3 levels, we totally agree that just using results based on H1c/d/e triple knockout mESCs to support our conclusion is not sufficient. Hence, we decided to do ChOR-seq in a condition where H1 levels vary while H3K27me3 levels remain largely the same. From response Fig5.A and B, we see that the bulk levels of H3K27me3 do not significantly changed during ~10 hrs IAA treatment, but the H1c/e reduced by ~40%. Hence, we labeled newly synthesized DNA by EdU pulsing after 6 hrs of IAA treatment (T0) and traced the restoration of H3K27me3 in the following 4 hrs. This setup allows us to analyze the contribution of H1 in promoting H3K27me3 restoration post-replication. However, if we use 12 hrs +/- auxin as the T0 time point, this setup is similar to the WT and H1-TKO asynchronised ChOR-seq as the H3K27me3 level is different in two groups during EdU pulsing and H3K27me3 restoration.
- (2) We have performed the semi-quantitative western blot analysis using samples in the H1-AID setup showed in the revised manuscript as Fig.5B and Fig.S5I (Response Figure.5A and 5B). The result shows that the bulk levels of H3K27me3 do not significantly changed during 10 hrs treatment with IAA, while H1 almost reduced by ~40% at the endpoint. This suggests that differences in H1 levels, but not total H3K27me3 levels, account for differences in H3K27me3 restoration rate.
- (3) We analyzed H3K27me3 enrichment at T1, T2 and T4 (Response Figure.5C) and found that for all time points we analyzed, differences between -IAA and +IAA are statistically significant for all time points including T1, T2 and T4. However, the referee raised a very interesting question concerning the dynamics of H1 during the process of chromatin maturation. However, there is no commercially available ChIP-grade pan-H1 antibody for mammalian cells to track the restoration of H1 post-replication. Nevertheless, as a follow-up work, we are currently characterizing the recycling kinetics and the function of histone

H1 shortly after DNA replication in *Drosophila* cells (*Drosophila* has only one H1 gene).
 (4) We have replicated the ChOR-seq experiments in HA-AID-H1c/e cell lines and shown the two replicates in Response Figure.4B.

Response Figure.5

6. One comment in the first revision was that the manuscript generally had a problem with not referencing published work when the authors report similar findings. This has been improved, except in a few places:

a. It is still unclear what is novel in figure 6 compared to published work. The authors present their data and conclusions in a long paragraph and then at the end state: 'Taken together, our results validated previous findings^{23,37}'. The studies should be cited at the appropriate places, and it should be highlighting what is novel – alternatively the authors should consider to move the data to the supplements if they mainly confirm published literature. Please also include a reference to Willcockson et al., 2021 Nature (Extended data fig. 7).

b. The authors removed most claims of causality with respect to restoration rate and gene expression, but should correct this in the discussion: 'Importantly, we also found that the restoration rate of H3K27me3 is negatively correlated with the mRNA levels of the genes among tissue-specific, bivalent, ES and house-keeping gene clusters, which ensures the proper gene expression program in mESCs.'

And title on supplementary figure 2: 'Supplementary figure 2. The restoration rate of H3K27me3 is pivotal for the post-replication gene silencing.'

Response: We thank the referee for the suggestion.

(a) Although several published studies have reported H1 could stimulate PRC2 activity, the exact stimulating mechanism is still unclear. In summary, there are 3 possibilities regarding the molecular basis of this stimulation activity, including 1) H1-mediated chromatin compaction; 2) H1-dependent PRC2 recruitment; 3) H1-induced allosteric effects. In 2006, Cyrus Martin, et al. reported that the PRC2 has substrate preferences toward H1-enriched chromatin. In 2012, Wen Yuan et al. reported that PRC2 activity is regulated by the density of its substrate nucleosome arrays through the interaction between PRC2 complex and a fragment of the H3 N terminus. Recently, Willcockson et al. reconstituted di-nucleosome with either the C-terminal truncated H1 or a nucleosome-binding-deficient H1 and performed HMT assay. In this study, they were not able to exclude the contribution of potential H1-mediated PRC2 recruitment or allosteric effects by H1. In our work we reconstituted different chromatin substrates to perform *in vitro* HMT assays and established targeting system at cellular level. To exclude any H1-

mediated PRC2 recruitment or allosteric effects, we designed our experiment accordingly and found that H1 does not affect PRC2 activity on mono-nucleosomes. Moreover, using 12×nucleosome ligation system, we can get direct observation of the propagation of H3K27me3 along chromatin fiber facilitated by H1. Besides, our ChIP-seq results in targeting system showed that targeting HA-PRC2-TetR efficiently deposits H3K27me3 across 8×TetO sites widely spreading around in wild type cells, however in H1-TKO cells the peak height of H3K27me3 is 10× lower and the spreading width is also narrowed. Together, we concluded that the propagation of H3K27me3 facilitated by H1-mediated chromatin compaction is a very important mechanism underlying the fast restoration of H3K27me3 after DNA replication. Therefore, we considered these data as an important part of the manuscript and did not move them to the supplements. Besides, we corrected the reference citation of Figure 6 in the revised manuscript on Page 11.

- (b) We have corrected the discussion according to the suggestion in the revised manuscript as “Importantly, we also found that the restoration rate of H3K27me3 is negatively correlated with the mRNA levels of the genes among tissue-specific, bivalent, ES and house-keeping gene clusters, which might have a potential role in the post-replication gene silencing” on Page 13 and corrected the title of supplementary Figure 2 as “The restoration rate of H3K27me3 is positively correlated with repressive chromatin states” in the revised SI on Page 4.

REVIEWERS' COMMENTS

Reviewer #1 (Remarks to the Author):

The authors have addressed all my comments comprehensively and I fully support publication of their exciting work.